# Rec-R1: Bridging Generative Large Language Models and User-Centric Recommendation Systems via Reinforcement Learning

**Jiacheng Lin**  *jl254@illinois.edu*
*University of Illinois Urbana-Champaign*

**Tian Wang**  *Shtimwang@gmail.com*
*Independent Researcher*

**Kun Qian**  *kunqian699@gmail.com*
*Independent Researcher*

**Reviewed on OpenReview:** *https://openreview.net/forum?id=YBRU9MV2vE*

## Abstract

We propose REC-R1, a general reinforcement learning framework that bridges large language models (LLMs) with recommendation systems through closed-loop optimization. Unlike prompting and supervised fine-tuning (SFT), REC-R1 directly optimizes LLM generation using feedback from a fixed, black-box recommendation model—without relying on synthetic SFT data from proprietary models like GPT-4o. This avoids the substantial cost and effort required for data distillation. To verify the effectiveness of REC-R1, we evaluate REC-R1 on three representative tasks: product search, sequential recommendation, and product re-ranking. Experimental results demonstrate that REC-R1 not only consistently outperforms prompting- and SFT-based methods, but also achieves remarkable gains over strong discriminative baselines, even when used with simple retrievers like BM25. More impressively, REC-R1 preserves the general-purpose capabilities of the LLM, in contrast to SFT, which often impairs instruction-following and reasoning. These findings suggest REC-R1 as a promising foundation for continual task-specific adaptation without catastrophic forgetting.

## 1 Introduction

Recommendation systems (RecSys) have become essential components in various real-world applications, from e-commerce (Schafer et al., 1999; Valencia-Arias et al., 2024) and video platforms (Lubos et al., 2023; Covington et al., 2016) to news delivery (Raza & Ding, 2022; Wu et al., 2023) and social media (Campana & Delmastro, 2017). Despite the remarkable progress of RecSys over the decades, modern systems still face fundamental limitations. Most notably, they lack open-domain world knowledge and struggle to understand users' underlying motivations and preferences (Lin et al., 2025). These shortcomings often lead to suboptimal recommendation performance, especially in complex scenarios such as when user intent is implicit or expressed in natural language (Hou et al., 2024a; He et al., 2023).

Recent advances in generative large language models (LLMs) have opened new possibilities for enhancing recommendation systems (Xi et al., 2024; Bao et al., 2023). Trained on massive web-scale corpora, LLMs possess extensive open-world knowledge, alongside strong capabilities in natural language understanding and reasoning (Achiam et al., 2023; Grattafiori et al., 2024; Yang et al., 2024; Lin et al., 2024). These strengths make LLMs particularly suitable for user-centric recommendation scenarios, where the key challenge is to understand language-driven inputs, capture implicit user intent, and align recommendations with user needs.

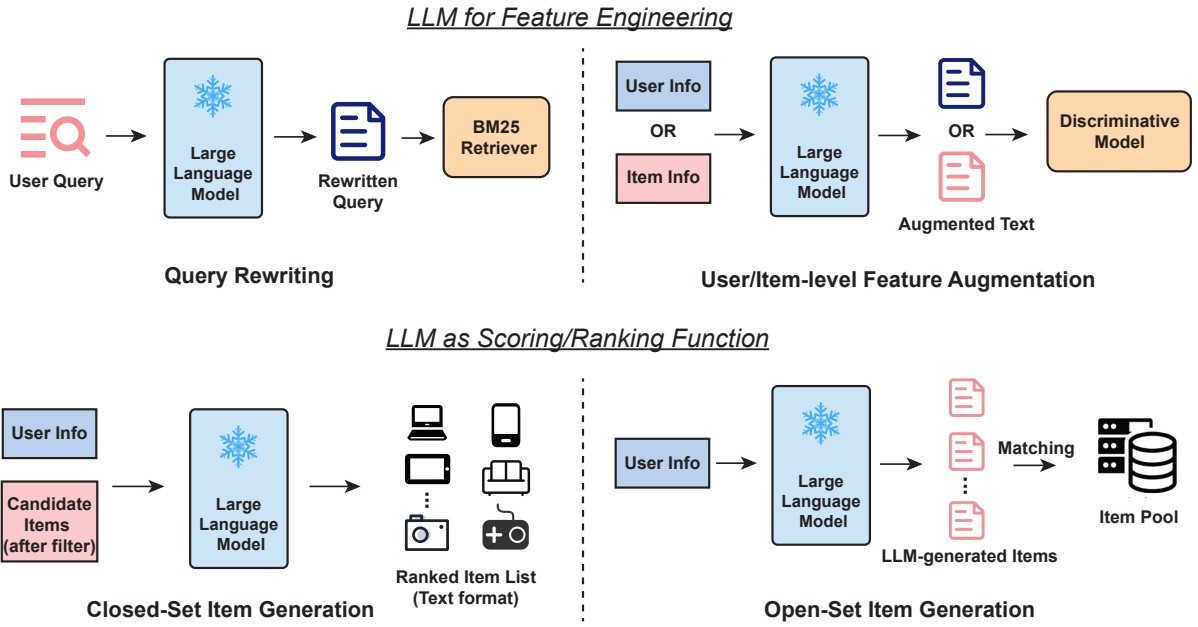

Figure 1: Illustration of how generative LLMs are applied in recommender systems (LLM4Rec), following the taxonomy in Lin et al. (2025). The upper row shows the use of LLMs for feature engineering, including (1) Query Rewriting, where the LLM reformulates the input query to improve retrieval, and (2) User/Item-level Feature Augmentation, where the LLM encodes user or item information into richer textual representations as input to a downstream model. The lower row demonstrates the use of LLMs as Scoring/Ranking Functions, including (3) Closed-Set Item Generation, where the LLM ranks a given candidate list, and (4) Open-Set Item Generation, where the LLM directly generate candidate items and matches them to a product pool. Note that this figure primarily reflects the inference-time setting—thus all LLMs are frozen. **Our proposed Rec-R1 is compatible with all paradigms shown here** (see Appendix A).

As a result, recent studies have explored using LLMs in various stages of the recommendation pipeline, including query rewriting (Peng et al., 2024; Li et al., 2023b), user intent summarization (Torbati et al., 2023), and so on, to improving the performance of downstream recommendation tasks such as retrieval and ranking. These methods typically employ either zero- or few-shot prompting (Xi et al., 2024; Lyu et al., 2023; Ren et al., 2024; Li et al., 2023b) or supervised fine-tuning (SFT) (Luo et al., 2024b; Li et al., 2023a; Yang et al., 2023; Liao et al., 2024; Ji et al., 2024) to adapt LLMs to recommendation tasks. Figure 1 illustrates the major paradigms where LLMs are applied in RecSys.

However, most existing approaches still treat LLMs and recommendation models as disjoint components, with **no closed feedback loop between LLM generation and recommendation performance** (Peng et al., 2024; Zheng et al., 2023; Luo et al., 2024b; Li et al., 2023a; Yang et al., 2023; Liao et al., 2024). As a result, LLMs are typically optimized using proxy objectives rather than being directly trained using feedback from RecSys, which is often inconsistent with the ultimate goal of improving recommendation quality. Moreover, constructing high-quality supervision data for intermediate tasks—such as query rewriting—typically requires either human annotation, LLM APIs (e.g., GPT-4), or mining from historical interaction logs (Peng et al., 2024; Hou et al., 2024a). Nonetheless, these sources rarely produce data that is truly aligned with optimal recommendation performance. Worse still, generating such data at scale is both time-consuming and financially expensive, especially when relying on human annotation or commercial LLMs. Figure 2 presents our proof-of-concept comparison that highlights the limitations of SFT in terms of both effectiveness and overhead.

To address these challenges, we propose REC-R1, a general framework that leverages reinforcement learning (RL) to bridge generative LLMs and downstream black-box RecSys through closed-loop optimization. In

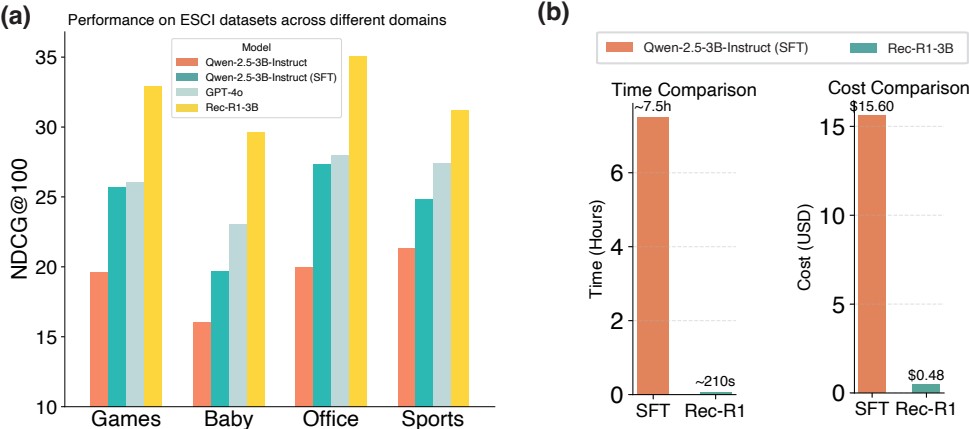

Figure 2: **Proof-of-concept comparison under a small-scale setup, illustrating the limitations of SFT based on GPT-4o-generated data. (a)** Performance on the ESCI dataset. The SFT baseline fine-tunes using data generated by GPT-4o, and its performance is inherently upper-bounded by the performance of GPT-4o itself. **(b)** Comparison of training time and cost. The total time for SFT includes both the data generation phase using GPT-4o and subsequent model fine-tuning. REC-R1 requires no additional data generation, and we report the minimal training time and cost required to match the performance of the SFT and GPT-4o model. See Appendix D for cost estimation details.

contrast to existing approaches that rely on static data and proxy supervision, REC-R1 enables LLMs to learn directly from recommendation feedback—such as retrieval or ranking metrics—thus aligning the generation process with the ultimate goal of improving recommendation quality. Specifically, given any recommendation-relevant input, such as a user query or behavioral history, an LLM generates a textual output that is consumed by a downstream recommendation model. This textual output varies in form depending on the task, such as a rewritten query, a synthesized user profile, or an enriched textual description of an item. The recommendation system then evaluates the quality of the LLM-generated text using rule-based performance metrics (e.g., NDCG, Recall), which are transformed into reward signals for optimizing the LLM via RL. Through repeated interaction with the recommendation system, the LLM gradually learns to generate inputs that are better aligned with the system's objectives, thereby improving recommendation performance without relying on suboptimal intermediate supervision.

REC-R1 is model-agnostic and task-flexible: it can be integrated with a wide range of recommendation architectures—including sparse retrievers (e.g., BM25), dense discriminative models, and hybrid pipelines—without requiring any modifications to their internal structures. It also supports diverse generation tasks as long as the generated text can be consumed by the downstream recommendation system. Moreover, since REC-R1 relies solely on black-box feedback in the form of recommendation performance metrics, it does not require access to model gradients or internal parameters, making it easy to deploy on top of existing production systems. It also eliminates the need for constructing SFT data, allowing the generative model to be optimized directly through interactions.

We evaluate REC-R1 in three representative recommendation scenarios—product search, sequential recommendation, and product re-ranking—to demonstrate its effectiveness, though the framework itself is broadly applicable to a wider range of recommendation tasks (see Appendix A). In product search, we observe that applying the REC-R1 framework significantly improves overall recommendation performance, achieving state-of-the-art results on the evaluated benchmarks. In the sequential recommendation setting, REC-R1 leads to consistent gains. More importantly, it shows strong performance in cold-start scenarios, where user profile information is absent, outperforming widely used sequential baselines. In product re-ranking, REC-R1 also shows clear superiority over both strong cross-encoder and LLM-based reranker baselines. Beyond performance gains, we further investigate whether REC-R1 preserves the general capabilities of the underlying language model. On the IFEval benchmark (Zhou et al., 2023), which measures instruction following capabilities, REC-R1 maintains or even improves performance, while SFT causes a drop of over 27 points.

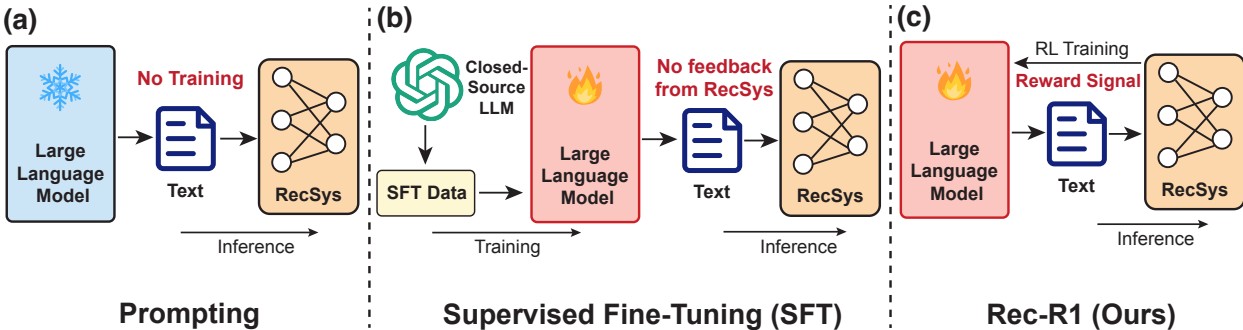

Figure 3: **Comparison of three paradigms for using LLMs in recommendation systems. (a)** Prompting uses a frozen LLM to generate textual inputs for the recommendation system, without any model updates. **(b)** SFT trains the LLM to imitate outputs generated by a stronger model (e.g., GPT-4o), but the training process does not involve any RecSys feedback. **(c)** REC-R1 introduces a closed-loop RL framework, where the LLM is optimized directly using reward signals from the recommendation system, without requiring external annotation or data distillation. Unlike SFT, which relies on labeled intermediate outputs (e.g., rewritten queries) from closed-source models, REC-R1 operates directly on the same data and learns from the recommendation performance.

This suggests that REC-R1 enables task-specific adaptation without compromising the general-purpose capabilities of the initialized LLM. We summarize our main contributions as follows:

- We propose REC-R1, a general reinforcement learning framework that bridges generative LLMs and recommendation systems through reward-driven optimization. Our approach is model-agnostic with respect to the recommendation system and supports diverse tasks.

- We conduct extensive experiments on three representative recommendation tasks, i.e., product search, sequential recommendation and product re-ranking, demonstrating that REC-R1 significantly improves performance across different recommendation architectures. Notably, in product search, REC-R1 improves the NDCG@100 score by up to 21.45 points for BM25-based retrievers, and by up to 18.76 points for dense discriminative models, compared to their respective baselines.

- REC-R1 preserves the general-purpose capabilities of the initialized LLM while achieving strong task-specific performance, outperforming supervised fine-tuning in both recommendation effectiveness and instruction-following generalization.

## 2 Rec-R1

### 2.1 Problem Formulation

We begin by modeling how LLMs are integrated into RecSys. In this general setup, the LLM receives an input $s \in \mathcal{S}$, which may represent a user query, behavioral history, or contextual information. The LLM then generates a textual output $a \in \mathcal{A}$, such as a rewritten query, an enriched item description, or a synthesized user profile. This output is consumed by a downstream recommendation model, which produces a performance-based evaluation $f(a|s) \in \mathbb{R}$, such as NDCG, Recall, or any task-specific metric.

The behavior of the LLM is governed by a conditional generation policy $\pi_\theta(a|s)$ where $\theta$ denotes the parameters of the generative LLM. The objective is to find a policy that maximizes expected recommendation performance:

$$\max_\theta \mathbb{E}_{s \sim p(s), a \sim \pi_\theta(a|s)}[f(a|s)] \tag{1}$$

Here, $p(s)$ denotes the empirical distribution over recommendation-relevant inputs provided to the LLM.

## 2.2 Theoretical Analysis of Existing Paradigms' Limitations

While the goal in recommendation-oriented LLM usage is to maximize downstream performance $\mathbb{E}[f(a|s)]$, existing paradigms fail to optimize this objective directly.

Prompting-based methods, including zero-shot and few-shot prompting, treat the LLM as a frozen generator. These approaches rely on manually constructed prompts or few examples to elicit desirable outputs. However, since the model parameters $\theta$ are not updated, the policy $\pi_\theta(a|s)$ remains fixed and cannot adapt to task-specific feedback, resulting in suboptimal recommendation performance.

Supervised fine-tuning forces the LLM to imitate outputs generated by a stronger model, such as GPT-4o. Formally, this corresponds to maximizing the log-likelihood of actions sampled from the data-generating policy $\pi_g$ under the learned policy $\pi_\theta$:

$$\max_\theta \mathbb{E}_{s\sim p(s), a\sim \pi_g(a|s)}[\log \pi_\theta(a|s)] \tag{2}$$

This maximum likelihood estimation (MLE) objective encourages the learned policy $\pi_\theta$ to imitate $\pi_g$, but does not consider the downstream performance $f(a|s)$ during optimization. We now show that this training procedure imposes a fundamental performance ceiling, which we formalize in the following fact.

**Fact 1** (SFT Converges Toward the Data-Generating Policy). *Let $\pi_g(a|s)$ be a fixed data-generating policy, and consider the supervised fine-tuning (SFT) objective:*

$$\pi_{\theta^*} = \arg\max_\theta \mathbb{E}_{s\sim p(s), a\sim \pi_g(a|s)}[\log \pi_\theta(a|s)]. \tag{3}$$

*Assume:*

*(i)* ***(Sufficient Expressivity)*** *The policy class $\{\pi_\theta(\cdot|s)\}$ is expressive enough to closely approximate the data-generating policy $\pi_g(\cdot|s)$, i.e., $\inf_\theta \mathbb{E}_{s\sim p(s)}[D_{\mathrm{KL}}(\pi_g(\cdot|s)\|\pi_\theta(\cdot|s))] = 0$.*

*(ii)* ***(Optimization Convergence)*** *The optimization process converges to a global maximum of the MLE objective.*

*(iii)* ***(Data Sufficiency)*** *Data-generating policy generates a sufficiently large amount of training samples, so the empirical distribution $\hat{p}(s,a)$ accurately approximates the true distribution $p(s)\pi_g(a|s)$, i.e.,*

$$\hat{p}(s,a) \xrightarrow{a.s.} p(s)\pi_g(a|s) \quad as \quad N \to \infty,$$

*where $N$ is the number of training samples generated.*

*Then the optimal policy $\pi_{\theta^*}$ is the one that minimizes the KL divergence to the data-generating policy $\pi_g$:*

$$\pi_{\theta^*} = \arg\min_\theta \mathbb{E}_{s\sim p(s)}\left[D_{\mathrm{KL}}(\pi_g(\cdot|s)\|\pi_\theta(\cdot|s))\right]. \tag{4}$$

The proof can be found in Appendix C.1. Fact 1 reveals a fundamental limitation of supervised fine-tuning: the learned policy $\pi_{\theta^*}$ is inherently constrained to imitate the data-generating policy $\pi_g$. Consequently, the recommendation performance of an SFT-trained model can at best approach—but never exceed—the performance of the policy used to generate the training data (e.g., GPT-4o). This fact is empirically supported by our experiments on the ESCI dataset, as illustrated in Figure 2(a), where the performance of the SFT-trained model closely matches but does not surpass GPT-4o. However, as GPT-4o itself is not explicitly optimized for the downstream recommendation task, its performance is typically suboptimal.

## 2.3 The Rec-R1 Framework

To overcome the limitations of prompting and SFT, we introduce REC-R1, a general framework that bridges generative LLMs and recommendation systems through reinforcement learning. Rather than imitating a static data-generating policy, REC-R1 directly optimizes the LLM policy $\pi_\theta$ based on feedback from the

downstream recommender—thereby aligning the generation process with the true objective: maximizing recommendation performance. See Figure 3 for a comparison of these paradigms.

At its core, REC-R1 casts the LLM-RecSys interaction as a closed-loop optimization process, where the LLM produces text (e.g., rewritten queries, user profiles, or item descriptions), and the recommendation model evaluates the result using task-specific metrics. These evaluation scores are then transformed into scalar reward signals for policy optimization via reinforcement learning.

A key strength of REC-R1 lies in its ability to optimize the LLM using direct feedback from the recommendation system. This feedback is formalized as a scalar reward $r = f(a|s) \in \mathbb{R}$, which quantifies how well the LLM-generated output $a$ performs in the downstream task given input $s$. The reward can be instantiated using any differentiable or non-differentiable metric that reflects recommendation quality, such as NDCG@K and Recall@K. Formally, the optimization objective is to find a generation policy $\pi_\theta(a|s)$ that maximizes the expected reward:

$$\max_\theta \mathbb{E}_{s \sim p(s),\, a \sim \pi_\theta(a|s)} \left[ f(a|s) \right]. \tag{5}$$

Unlike SFT, this objective does not rely on manually labeled supervision or imitation of a fixed policy. Instead, it allows the model to continuously adapt its behavior to maximize performance on the downstream recommendation tasks. From the perspective of the optimization objective, Eq. 2 minimizes a proxy loss that indirectly relates to recommendation quality, whereas Eq. 5 directly optimizes the true recommendation objective, ensuring alignment with the downstream evaluation metrics. Following DeepSeek-R1 (Guo et al., 2025), we adopt Group Relative Policy Optimization (GRPO) (Shao et al., 2024) to optimize the LLM policy. Compared to traditional algorithms such as PPO (Schulman et al., 2017), GRPO significantly reduces memory consumption during training while maintaining competitive performance. Further, we use rule-based reward functions derived from standard evaluation metrics (e.g., NDCG, Recall) rather than training a separate reward model, which helps mitigate reward hacking and avoids introducing additional biases.

## 3 Experiments

To validate the effectiveness of REC-R1, we conduct experiments on three representative recommendation scenarios: product search (§3.1), sequential recommendation (§3.2), and product re-ranking (§3.3). We also perform detailed analyses to examine the generalization ability after training (§3.4). More discussions and analysis can be found in Appendix F and case study in Appendix G.

### 3.1 Product Search

#### 3.1.1 Experimental Setup

**Task Definition.** In the product search task, the user provides a natural language query $s \in \mathcal{S}$, which expresses an information need (e.g., "a waterproof camera for hiking"). The goal of the recommendation system is to retrieve a ranked list of items that best match this query. To improve retrieval quality, the LLM generates a textual transformation $a \in \mathcal{A}$—such as a rewritten or clarified version of the query—which is then fed into a downstream retriever.

The retriever returns a ranked list of candidate items based on the textual input $a$, and the system evaluates performance using a relevance dictionary $\mathcal{D}$ that maps each original input $s$ to its corresponding ground-truth item list. The reward score $f(a|s) \in \mathbb{R}$ is computed by comparing the retrieved list (from $a$) against the target set $\mathcal{D}(s)$ associated with the original query $s$. We use NDCG@100 as the evaluation metric, which captures both relevance and ranking position. This reward function serves as the feedback signal in REC-R1 to optimize the LLM's generation policy $\pi_\theta(a|s)$. While $\mathcal{D}$ is derived from the original datasets in our experiments, in real-world deployments it can be constructed from various sources such as most recent user interaction logs or click-through data, making REC-R1 promising to production-scale recommendation systems.

Table 1: **Performance comparison of different methods on conventional product search (ESCI) tasks.** We report the NDCG@100 scores. The best performance score is denoted in **bold**, with the second and third best __underlined__. The numbers in gray indicate the absolute improvement of REC-R1 over their corresponding base retrievers (BM25 or BLAIR).

| Model | Video Games | Baby Products | Office Products | Sports and Outdoors |
|---|---|---|---|---|
| **Sparse retrieval baselines** | | | | |
| BM25 | 12.44 | 15.12 | 23.96 | 19.48 |
| GPT-4o$_{+\text{BM25}}$ | 26.06 | 23.05 | 27.98 | 27.38 |
| Qwen-2.5-3B-Instruct$_{+\text{BM25}}$ | 19.63 | 16.03 | 19.96 | 21.36 |
| **Dense retrieval baselines** | | | | |
| RoBERTa$_{\text{BASE}}$ | 0.16 | 0.00 | 0.00 | 0.17 |
| SimCSE$_{\text{BASE}}$ | 2.21 | 5.68 | 8.58 | 8.03 |
| BLAIR$_{\text{BASE}}$ | 9.75 | 15.20 | 17.19 | 17.08 |
| RoBERTa$_{\text{LARGE}}$ | 0.00 | 0.00 | 0.00 | 0.00 |
| SimCSE$_{\text{LARGE}}$ | 6.59 | 9.71 | 13.63 | 11.90 |
| BLAIR$_{\text{LARGE}}$ | 15.88 | 15.96 | 21.17 | 18.30 |
| GPT-4o$_{+\text{BLAIR-BASE}}$ | 20.13 | 24.57 | 21.83 | 22.97 |
| GPT-4o$_{+\text{BLAIR-LARGE}}$ | 23.99 | 24.10 | 22.99 | 24.67 |
| Qwen-2.5-3B-Instruct$_{+\text{BLAIR-BASE}}$ | 10.56 | 16.23 | 13.61 | 17.09 |
| Qwen-2.5-3B-Instruct$_{+\text{BLAIR-LARGE}}$ | 17.34 | 16.10 | 17.29 | 17.74 |
| **Ours** | | | | |
| REC-R1-3B$_{+\text{BM25}}$ | **33.89** | **29.27** | **34.61** | __31.92__ |
| | (+21.45) | (+14.15) | (+10.65) | (+12.44) |
| REC-R1-3B$_{+\text{BLAIR-BASE}}$ | __28.51__ | __29.24__ | __33.98__ | __30.71__ |
| | (+18.76) | (+14.04) | (+16.79) | (+13.63) |
| REC-R1-3B$_{+\text{BLAIR-LARGE}}$ | __31.41__ | __28.76__ | __34.12__ | **32.49** |
| | (+15.53) | (+12.80) | (+12.95) | (+14.19) |

**Datasets.** We consider two distinct settings for product search: (1) **conventional product search**, where the input query is a short phrase or keyword-based expression (e.g., "noise-canceling headphones"), and (2) **complex product search**, where the input is a rich and long natural language context, often involving implicit preferences or use-case scenarios. To evaluate these two settings, we adopt two datasets: the **ESCI** dataset for conventional product search (Reddy et al., 2022) and the **Amazon-C4** dataset for complex product search (Hou et al., 2024a). More details can be found in Appendix E.1.1. For all experiments in this paper, we report test performance based on the checkpoint that achieves the best validation score.

**Baselines.** We compare REC-R1 against a range of baselines. For sparse retrieval, we use the BM25, as well as prompting-enhanced variants where a frozen LLM (GPT-4o or Qwen-2.5-3B-Instruct (Yang et al., 2024)) rewrites the input query before retrieval. For dense retrieval, we include discriminative models such as RoBERTa (Liu et al., 2019), SimCSE (Gao et al., 2021), and BLAIR (Hou et al., 2024a), with and without prompting-based query rewriting. In contrast, REC-R1 starts from the Qwen-2.5-3B-Instruct model and is trained via RL to generate rewritten queries that directly optimize retrieval performance.

### 3.1.2 Results

**Results on ESCI.** Table 1 reports NDCG@100 scores on the conventional product search benchmark ESCI. We observe that REC-R1 consistently improves retrieval performance across all four domains and retriever architectures. Notably, even when applied to the sparse BM25 retriever, REC-R1 yields substantial gains—up to +21.45 NDCG points in the Video Games domain—demonstrating its ability to enhance classic lexical systems. For dense retrievers such as BLAIR, REC-R1 brings improvements of up to +18.76 over the base models, and consistently outperforms prompting-based rewriting with GPT-4o. Remarkably, REC-R1 achieves the best performance across all four product categories, underscoring its effectiveness and overall superiority.

**Results on Amazon-C4.** We further evaluate REC-R1 on the Amazon-C4 dataset, which contains complex product search queries expressed in natural language. This setting also enables us to assess the model's cross-domain generalization ability: we train on queries from all categories *except* the four test domains (Video

Table 2: **Performance comparison of different methods on complex product search (Amazon-C4) tasks.** We report the NDCG@100 scores. The best performance score is denoted in **bold**, with the second and third best underlined. The numbers in gray indicate the improvement of REC-R1 over their corresponding base retrievers (BM25 or BLAIR).

| Model | Video Games | Baby Products | Office Products | Sports and Outdoors |
|---|---|---|---|---|
| **Sparse retrieval baselines** | | | | |
| BM25 | 7.82 | 6.39 | 8.70 | 7.28 |
| GPT-4o$_{+\text{BM25}}$ | 13.02 | 12.45 | 12.64 | 11.59 |
| Qwen-2.5-3B-Instruct$_{+\text{BM25}}$ | 10.33 | 8.62 | 9.61 | 9.05 |
| **Dense retrieval baselines** | | | | |
| RoBERTa$_{\text{BASE}}$ | 0.22 | 0.12 | 0.33 | 0.13 |
| SimCSE$_{\text{BASE}}$ | 6.05 | 6.22 | 3.71 | 4.33 |
| BLAIR$_{\text{BASE}}$ | 19.14 | 19.53 | 17.43 | 20.02 |
| RoBERTa$_{\text{LARGE}}$ | 0.00 | 0.06 | 0.00 | 0.00 |
| SimCSE$_{\text{LARGE}}$ | 5.03 | 7.43 | 5.41 | 8.12 |
| BLAIR$_{\text{LARGE}}$ | 24.86 | 22.44 | 18.92 | 24.54 |
| GPT-4o$_{+\text{BLAIR-BASE}}$ | 21.12 | 19.54 | 17.22 | 22.68 |
| GPT-4o$_{+\text{BLAIR-LARGE}}$ | 23.40 | 21.00 | 18.69 | 25.74 |
| Qwen-2.5-3B-Instruct$_{+\text{BLAIR-BASE}}$ | 15.82 | 18.07 | 14.34 | 16.96 |
| Qwen-2.5-3B-Instruct$_{+\text{BLAIR-LARGE}}$ | 18.20 | 19.19 | 15.37 | 18.94 |
| **Ours** | | | | |
| REC-R1-3B$_{+\text{BM25}}$ | 18.91 | 20.55 | 19.24 | 20.06 |
| | (+11.09) | (+14.16) | (+10.54) | (+12.78) |
| REC-R1-3B$_{+\text{BLAIR-BASE}}$ | 21.69 | 25.62 | 22.17 | 24.22 |
| | (+2.82) | (+6.09) | (+4.74) | (+4.20) |
| REC-R1-3B$_{+\text{BLAIR-LARGE}}$ | **26.51** | **27.04** | **23.10** | **27.40** |
| | (+1.65) | (+4.60) | (+4.18) | (+2.86) |

Games, Baby Products, Office Products, and Sports and Outdoors), and evaluate performance on these held-out domains.

As shown in Table 2, REC-R1 achieves the best performance across all four domains and retriever architectures, demonstrating strong generalization beyond the training distribution. This cross-domain evaluation highlights REC-R1's ability to generalize from training on one set of product categories to effectively handling unseen categories during testing. Moreover, prompting-based query rewriting using frozen LLMs (e.g., GPT-4o or Qwen-Instruct) either yields negligible improvement or even degrades performance—particularly on dense retrievers. In contrast, both BM25 and dense models see notable gains when combined with REC-R1, indicating the value of interaction-based learning. By receiving direct feedback from the recommendation system during training, the LLM gradually learns how to rewrite queries in a way that maximizes downstream task performance.

Note that we do not report SFT results in our tables, as prior analysis (Theorem 1) shows that, under sufficient data and optimization, the learned policy from SFT converges toward the data-generating model (e.g., GPT-4o). Therefore, we use GPT-4o performance as a practical reference point. Across both ESCI and Amazon-C4, REC-R1 consistently outperforms GPT-4o-based prompting methods—demonstrating its potential to go beyond the limitations of the SFT paradigm in both performance and adaptability.

## 3.2 Sequential Recommendation

### 3.2.1 Experimental Setup

**Task Definition.** In this task, the model receives a user's historical interaction sequence $s \in \mathcal{S}$ (e.g., a list of previously viewed or purchased items) and is expected to recommend the most relevant next item. To support this process, the LLM generates a text $a \in \mathcal{A}$—a query describing what the user probably will purchase next. This could take the form of key attributes, product type, or usage scenario, serving as a query-like signal to input the downstream retriever. To evaluate performance, we define a relevance dictionary $\mathcal{D}$ that maps each historical sequence $s$ to the ground-truth next item. The reward score $f(a|s) \in \mathbb{R}$ is computed by

Table 3: **Performance comparison of different methods on the sequential recommendation task on Amazon Beauty dataset.** We report the Recall@k (R) and NDCG@k (N) scores. The best performance score is denoted in **bold**. The numbers in gray indicate the absolute improvement of REC-R1 over the initialized policy, i.e., Qwen-2.5-3B-Instruct.

| Model | Transductive Setting | | | | Inductive Setting | | | |
|---|---|---|---|---|---|---|---|---|
| | R@10 | N@10 | R@50 | N@50 | R@10 | N@10 | R@50 | N@50 |
| **Text-aware SRec baselines** | | | | | | | | |
| SASRec$_{+\text{BLAIR-BASE}}$ | 3.72 | 1.55 | 7.81 | 2.44 | 0.20 | 0.90 | 1.40 | 0.35 |
| SASRec$_{+\text{BLAIR-LARGE}}$ | 3.90 | 2.12 | 8.74 | 3.17 | 0.40 | 0.15 | 1.50 | 0.40 |
| UniSRec$_{+\text{BLAIR-BASE}}$ | 4.09 | **2.23** | **8.74** | **3.21** | 3.70 | 2.08 | 6.20 | 2.61 |
| UniSRec$_{+\text{BLAIR-LARGE}}$ | **4.09** | 2.17 | 8.55 | 3.17 | 3.60 | 2.08 | 6.00 | 2.60 |
| **Query rewriting baselines** | | | | | | | | |
| GPT-4o$_{+\text{BM25}}$ | 0.00 | 0.00 | 1.48 | 0.33 | 1.00 | 0.50 | 2.90 | 0.90 |
| Qwen-2.5-3B-Instruct$_{+\text{BM25}}$ | 1.30 | 0.75 | 2.41 | 0.98 | 1.80 | 0.84 | 3.60 | 1.25 |
| **Ours** | | | | | | | | |
| REC-R1-3B$_{+\text{BM25}}$ | 3.53 | 1.74 | 5.76 | 2.22 | **6.00** | **3.38** | **8.30** | **3.89** |
| | (+2.23) | (+0.99) | (+3.35) | (+1.24) | (+4.20) | (+2.54) | (+4.70) | (+2.64) |

comparing the retrieved items (from $a$) against the target set $\mathcal{D}(s)$ using standard retrieval metrics such as Recall@K and NDCG@K. In our implementation, we use NDCG@K as the training reward for REC-R1. In real-world systems, the dictionary $\mathcal{D}$ can be constructed from user interaction logs, purchase sequences, or other behavioral data sources.

**Dataset.** We conduct experiments on the Amazon Beauty dataset following the split protocol from Hou et al. (2024a), where data are partitioned into training, validation, and test sets by absolute timestamp. To evaluate different generalization capabilities, we define two test-time settings: (1) **Transductive setting:** All candidate items in the test sequence (both history and target) have appeared in the training set; (2) **Inductive setting:** None of the test-time items are seen during training.

**Baselines.** We compare REC-R1 with two families of baselines: **(1)** *Text-aware Sequential Recommendation (SRec) models*, including SASRec (Kang & McAuley, 2018) and UniSRec (Hou et al., 2022), combined with BLAIR as the item encoder (base and large variants). These methods use sequence modeling with features from textual encoders. **(2)** *Prompting-based query rewriting*, where frozen LLMs (GPT-4o or Qwen-2.5-3B-Instruct) generate rewritten inputs from user histories, which are then fed into a retriever (e.g., BM25).

### 3.2.2 Results

Table 3 summarizes the performance under both transductive and inductive settings. We observe that the prompting-based baselines using frozen LLMs (e.g., GPT-4o and Qwen-2.5-3B-Instruct) perform poorly across the board—highlighting the difficulty of this task for generic LLMs without adaptation. However, when trained under the REC-R1 framework, the finetuned Qwen-2.5-3B-Instruct model exhibits significant performance gains, especially in the inductive setting. For instance, in Recall@10 and NDCG@50, REC-R1 improves upon its initialized policy by +4.20 and +2.64 points respectively. Moreover, REC-R1 outperforms strong sequential baselines like UniSRec in the inductive setting, demonstrating its superiority in cold-start or unseen-item scenarios.

In the transductive setting, REC-R1 remains competitive but lags behind specialized SRec models. This is not surprising, as traditional sequential recommendation methods are explicitly trained on large-scale user-item sequences and directly model sequential dependencies. In contrast, REC-R1 leverages LLMs to generate natural language queries about the user's next likely purchase—relying on their strength in reasoning and generalization. However, next-item prediction based solely on interaction history is often not a task that lends itself to explicit reasoning in language space. The relationship between past and future items may be weak or non-causal, making it inherently difficult for LLMs to perform this task effectively. More analysis

Table 4: **Performance comparison of different re-rankers on ESCI**. We report the NDCG@10 scores across four domains. The best performance is denoted in **bold**. The numbers in gray indicate the improvement of REC-R1 over its corresponding initialized model, i.e., Qwen2.5-3B-Instruct.

| Model | Video Games | Baby Products | Office Products | Sports and Outdoors |
|---|---|---|---|---|
| **Cross-encoder re-rankers** | | | | |
| Qwen3-Reranker-4B | 44.29 | 46.41 | 48.06 | 47.42 |
| Qwen3-Reranker-8B | 44.28 | 44.95 | 47.73 | 44.51 |
| **LLM-based re-ranker baselines** | | | | |
| Qwen2.5-3B-Instruct | 5.38 | 9.21 | 4.26 | 4.37 |
| GPT-4o | **61.57** | 56.18 | 53.62 | 53.62 |
| **Ours** | | | | |
| REC-R1-3B | 53.04 | **57.08** | **60.91** | **62.91** |
| | (+47.66) | (+47.87) | (+56.65) | (+58.54) |

of why REC-R1 performs better in the inductive setting than in the transductive setting can be found in Appendix E.2.4.

### 3.3 Product Re-ranking

#### 3.3.1 Experimental Setup

**Task Definition.** In the product re-ranking task, the input is a natural language query and an initial candidate list of $m$ items, denoted as $s = (q, C) \in \mathcal{S}$ where $q$ is the user query and $C = [i_1, \ldots, i_m]$ is the candidate set. The goal is to reorder $C$ so that items more relevant to $q$ are ranked higher. Given $s$, the LLM produces $a \in \mathcal{A}$, a re-ranked list of the $m$ candidates (i.e., a permutation of $C$), denoted by $a = [i_{\pi(1)}, \ldots, i_{\pi(m)}] \in \mathcal{A}$, where $\pi$ is a permutation function over $\{1, \ldots, m\}$. To evaluate performance, we define a relevance dictionary $\mathcal{D}$ that maps each input $s$ to corresponding ground-truth item list. The reward score $f(a|s) \in \mathbb{R}$ is computed by comparing the re-ranked list $a$ against the target item list $\mathcal{D}(s)$. We use NDCG@10 as the ranking metric.

**Dataset and Baselines.** We conduct experiments on the ESCI dataset (Reddy et al., 2022). The candidate items $C$ are retrieved using the Rec-R1-Retriever given the user query, and subsequently used for the re-ranking stage. To ensure fair comparison, all re-ranking models operate on the same candidate lists obtained from the Rec-R1-Retriever. We compare REC-R1 with two families of baselines: (1) *traditional cross-encoder re-rankers* (Zhang et al., 2025), where we adopt the state-of-the-art open-source Qwen3-Reranker family, including the 4B and 8B variants; and (2) *LLM-based re-rankers*, including GPT-4o and Qwen2.5-3B-Instruct, which directly generate ranking orders conditioned on the query and candidate list. Further implementation details and hyperparameters are provided in Appendix E.3.

#### 3.3.2 Results

Table 4 shows the re-ranking performance of different models on the ESCI dataset. From the table, REC-R1 consistently achieves the best overall performance across all domains. Notably, compared with the initialized Qwen2.5-3B-Instruct model, REC-R1 delivers very large improvements. Furthermore, against SoTA open-source embedding/reranker models (Qwen3-Reranker family), REC-R1 demonstrates substantial superiority. These results confirm that REC-R1 's optimization framework can also be applied to the closed-set item generation paradigm within LLM4Rec, further supporting our claim that Rec-R1 provides a general way to bridge LLMs and RecSys.

### 3.4 Does Rec-R1 Forget? A Generalization Analysis

While REC-R1 is explicitly designed to enhance recommendation performance, an important question is whether it can preserve the general-purpose capabilities of the underlying language model. To this end, we compare three models: (1) **Qwen-2.5-3B-Instruct**; (2) **Qwen-2.5-3B-Instruct (SFT)**, fine-tuned on

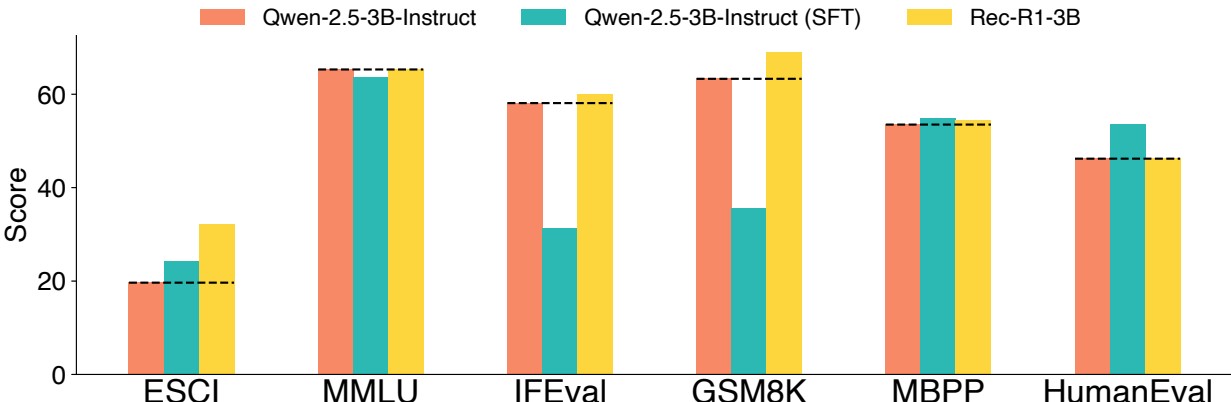

Figure 4: Generalization analysis across six benchmarks. We compare the initialized model (Qwen-2.5-3B-Instruct), its SFT variant trained on GPT-4o–generated SFT-data (ESCI), and our REC-R1-3B model trained via RL. **Note that Rec-R1 is only trained on the task-specific ESCI data, whose format drastically differs from the other benchmark datasets.**

ESCI query rewriting data generated by GPT-4o; and (3) **Rec-R1-3B**, trained using our RL framework on the same ESCI task, but without access to GPT-4o outputs. We evaluate all models across six tasks spanning different axes of generalization: ESCI (recommendation), MMLU (factual knowledge), IFEval (instruction following), GSM8K (math reasoning), and two coding benchmarks: MBPP and HumanEval. Results are shown in Figure 4. On ESCI, SFT yields improvements over the base model, while REC-R1 achieves substantially higher gains—without relying on GPT-4o-generated data. On MMLU, all models—including REC-R1 —achieve comparable accuracy. This suggests that neither SFT nor REC-R1 compromises the model's general knowledge.

However, striking differences emerge on IFEval. Here, SFT suffers a dramatic performance drop—losing over 27 points—**while Rec-R1 not only avoids degradation but actually improves slightly over the original model.** This highlights a key advantage of our REC-R1: by optimizing directly for task-specific performance without overriding the model's generative distribution via next token prediction, REC-R1 preserves instruction-following capabilities more effectively. We observe a similar trend on GSM8K, where REC-R1 improves upon the initialized model while SFT lags far behind. In contrast, both SFT and REC-R1 maintain strong performance on coding tasks like MBPP and HumanEval. This is likely because our SFT data involved JSON-style outputs to facilitate answer extraction, which do not interfere with the model's code generation ability. **These results highlight the promise of Rec-R1 as a new paradigm for LLM adaptation—one that enables strong task-specific improvements without compromising general capabilities.**

## 4 Conclusion

In this work, we present REC-R1, a reinforcement learning framework that bridge LLMs with recommendation systems through direct feedback. Rec-R1 achieves strong performance across tasks and retrievers while preserving general-purpose capabilities, offering a scalable alternative to prompting and SFT.

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

**Contents of Appendix**

## A   Applying Rec-R1 to Diverse Recommendation Paradigms

Figure 1 provides a taxonomy of how generative LLMs are incorporated into recommender systems, adapted from Lin et al. (2025). The taxonomy is divided into two main categories: **LLM for Feature Engineering** and **LLM as Scoring/Ranking Functions**, each with two sub-paradigms. In our experiments, we validate the applicability of Rec-R1 across all of the four major paradigms:

- **Query Rewriting:** In the product search task, we use BM25 as the retriever, and the LLM generates rewritten user queries to improve retrieval. This corresponds to the top-left panel of Figure 1 and demonstrates the of Rec-R1 for textual query rewriting.

- **User-Level Feature Augmentation:** When combined with dense discriminative models such as BLAIR, the LLM augments the input query with semantically richer expressions of user intent. This mirrors the top-right panel of Figure 1, where LLMs act as textual feature generators for downstream ranking models.

- **Open-Set Item Generation:** In the sequential recommendation task, the LLM is provided only with user-side information (e.g., purchase history), and is required to generate a textual prediction of the next item. This text is then matched against the item pool to select the final recommendation. This aligns with the bottom-right panel of Figure 1.

- **Closed-Set Item Generation:** In the product re-ranking task, the LLM is trained to rank or score candidates conditioned on user query and initialized product candidates through reward signals from ranking quality. This aligns with the bottom-left panel in Figure 1. Notably, prior work in this direction has largely relied on prompting (Zhang et al., 2023a; Hou et al., 2024b; Di Palma et al., 2023; Di Palma, 2023) or SFT (Yang et al., 2023; Luo et al., 2024b). However, as shown in our generalization experiments, such approaches may lead to catastrophic forgetting. In contrast, Rec-R1 maintains strong domain-specific performance without sacrificing general capabilities (§3.4), making it a more robust solution.

These results collectively highlight the versatility of Rec-R1 —it is applicable across diverse paradigms in recommender systems, and agnostic to the architecture of the retriever or scoring model.

**A Path Toward Unified Training.**   Given its compatibility across paradigms and its ability to retain general-purpose capabilities, we believe Rec-R1 provides a strong foundation for *continual, reinforcement-based alignment of LLMs with evolving recommendation goals*. Future work will explore extending Rec-R1 into more tasks, ultimately enabling lifelong recommendation agents that adapt flexibly to new tasks and domains without retraining from scratch.

## B   Related Work

### B.1   Generative LLMs for Recommendation Tasks

Recently, large language models (LLMs) have significantly impacted recommendation systems by leveraging their strong generalization, reasoning, and semantic understanding abilities. These approaches can be broadly categorized into several main directions.

**Feature Engineering and Augmentation.** LLMs have been extensively used to enrich recommendation data. They generate auxiliary features that enhance user profiling and item understanding, thus addressing data sparsity and improving the recommendation quality (Xi et al., 2024; Liu et al., 2025; Torbati et al., 2023; Shi et al., 2023; Li et al., 2024).

**LLM as Scoring and Ranking Functions.** Researchers have adapted LLMs as direct ranking or scoring components within recommendation pipelines. Methods such as P5 (Geng et al., 2022), M6-Rec (Cui et al., 2022), and InstructRec (Zhang et al., 2023b) explore LLMs' ability to simultaneously handle multiple recommendation subtasks, including scoring, generation, and re-ranking. Models like RecRanker (Luo

et al., 2024a) further leverage LLMs' natural language understanding to integrate multiple ranking strategies effectively.

**Conversational and Interactive Recommendations:** LLMs facilitate more sophisticated interactions between users and recommendation systems through conversational agents, significantly enhancing user engagement and recommendation explainability (Luo et al., 2024a; Zhou et al., 2020; Gao et al., 2023b).

For a comprehensive review of how recommender systems benefit from LLMs across different pipeline stages and application scenarios, readers can refer to the recent survey by Lin et al. (2025). Notably, our proposed REC-R1 framework is broadly applicable across these paradigms—it is not tailored to any specific application scenario or retriever architecture, but instead provides a general modeling and optimization approach for aligning LLMs with recommendation tasks via closed-loop learning.

### B.2   Reinforcement Learning for Recommendation Systems

Before the era of large language models (LLMs), reinforcement learning (RL) had been explored in recommendation systems for various objectives, such as optimizing long-term user satisfaction and improving sequential decision-making. These works typically reformulate recommendations as a Markov Decision Process (MDP), enabling agents to learn from user interaction sequences (Wang et al., 2020; Zhao et al., 2018). For example, Liu et al. (2023) extend DDPG to session-based recommendation, while Xin et al. (2020) propose self-supervised RL with SQN and SAC. Additional efforts incorporate negative sampling, contrastive learning, and reward modeling to enhance learning signals (Ren et al., 2023; Xin et al., 2022a;b). Different from these approaches, which aim to improve the recommendation model itself, our method treats the recommender as a black-box environment. We instead apply RL to optimize the LLM's generation policy, using feedback from the recommendation system to guide LLM training, which enables task-specific alignment without altering the underlying recommender.

In contrast, with the rise of LLMs, recent efforts begin to explore how to integrate RL and LLM to improve recommendation systems. For instance, Jeong et al. (2023) apply RLHF to align a language model with factuality, personalization, and appeal in movie recommendations. However, their framework uses a reward model trained offline, without interacting with the recommender system in the loop—mirroring the RLHF setup of InstructGPT. This approach not only lacks real-time adaptation to system feedback but also risks reward hacking. Other efforts take alternative views: Sun et al. (2024); Lu et al. (2024) attempt to bring in RecSys feedback but either restrict to DPO-style offline preference tuning (easily overfit on the static datasets) or narrow scenarios like sequential recommendation with fixed candidate sets.

Different from all of the above, our method REC-R1 directly optimizes the LLM with real-time reward signals from the recommendation system. The recommender is treated as a black-box environment, and the LLM adapts its generation policy (e.g., query rewriting or user profile generation) to maximize actual downstream task performance. This closed-loop RL training allows for general applicability across various recommendation tasks and retrievers, without relying on complex reward models or curated preference labels.

### B.3   LLMs for Query Rewriting

A growing body of research has investigated using large language models for query rewriting in retrieval tasks. In particular, recent works have explored leveraging LLMs to reformulate queries either in a zero-shot manner or via supervised fine-tuning (Gao et al., 2023a; Khattab & Zaharia, 2020; Ye et al., 2023; Mackie et al., 2023; Liu et al., 2022). These methods demonstrate that LLMs can generate semantically richer queries that better align with downstream retrieval models, thereby improving recall and ranking performance.

While these works provide important insights into query rewriting and optimization, there remain several key differences compared with our proposed Rec-R1 framework. First, closed-loop recommendation feedback: prior methods typically rely on synthetic or proxy reward signals, whereas we directly employ recommendation evaluation metrics (e.g., NDCG, Recall) from downstream models as reward signals—thus establishing a true closed-loop interaction between the LLM and the recommender system. Second, model scale and capability: most existing approaches were developed before the LLM era and used relatively weak models without strong reasoning or generalization abilities. In contrast, our work explicitly adapts powerful open-

domain LLMs to recommendation scenarios, addressing the capability gap of traditional RecSys. Finally, task scope: prior studies primarily focused on improving search query refinement, while our framework is designed to support a broader range of recommendation tasks.

## C Theorems and Proofs

### C.1 Proofs of Fact 1

**Lemma 1** (MLE Minimizes KL Divergence)**.** *Let $\pi_g(a|s)$ be a fixed target policy (e.g., the data-generating policy), and let $\pi_\theta(a|s)$ be a parameterized policy class. Consider the following maximum likelihood estimation (MLE) objective:*

$$\max_\theta \mathbb{E}_{s\sim p(s), a\sim \pi_g(a|s)}[\log \pi_\theta(a|s)].$$

*Then maximizing this objective with respect to $\theta$ is equivalent to minimizing the expected Kullback-Leibler (KL) divergence between the target policy $\pi_g$ and the parameterized policy $\pi_\theta$:*

$$\min_\theta \mathbb{E}_{s\sim p(s)} \left[ D_{\mathrm{KL}}\big(\pi_g(\cdot|s)\|\pi_\theta(\cdot|s)\big) \right].$$

*Proof.* We start by considering the expected negative log-likelihood under the distribution induced by $\pi_g$:

$$\mathbb{E}_{s\sim p(s), a\sim \pi_g(a|s)}[-\log \pi_\theta(a|s)] = \mathbb{E}_{s\sim p(s)}\mathbb{E}_{a\sim \pi_g(a|s)}[-\log \pi_\theta(a|s)] \qquad (6)$$

$$= \mathbb{E}_{s\sim p(s)} \left[ \sum_a \pi_g(a|s)\big( -\log \pi_\theta(a|s)\big) \right]. \qquad (7)$$

By definition of the KL divergence, we have the identity:

$$D_{\mathrm{KL}}\big(\pi_g(\cdot|s)\|\pi_\theta(\cdot|s)\big) = \sum_a \pi_g(a|s) \log \frac{\pi_g(a|s)}{\pi_\theta(a|s)}.$$

Rearranging terms gives:

$$\mathbb{E}_{a\sim \pi_g(a|s)}[-\log \pi_\theta(a|s)] = D_{\mathrm{KL}}\big(\pi_g(\cdot|s)\|\pi_\theta(\cdot|s)\big) + \mathbb{H}\big(\pi_g(\cdot|s)\big),$$

where $\mathbb{H}(\pi_g(\cdot|s)) = -\sum_a \pi_g(a|s) \log \pi_g(a|s)$ is the entropy of the fixed distribution $\pi_g(\cdot|s)$, which is independent of $\theta$.

Taking expectation over $s \sim p(s)$, we obtain:

$$\mathbb{E}_{s\sim p(s), a\sim \pi_g(a|s)}[-\log \pi_\theta(a|s)] = \mathbb{E}_{s\sim p(s)}[D_{\mathrm{KL}}(\pi_g(\cdot|s)\|\pi_\theta(\cdot|s))] + \mathbb{E}_{s\sim p(s)}[\mathbb{H}(\pi_g(\cdot|s))].$$

Since the second term is independent of $\theta$, minimizing the negative log-likelihood is equivalent to minimizing the KL divergence. Thus the lemma follows. $\square$

**Below is the proof of Fact 1.**

*Proof.* By Lemma 1, the supervised fine-tuning (SFT) objective of maximizing the expected log-likelihood:

$$\max_\theta \mathbb{E}_{s\sim p(s), a\sim \pi_g(a|s)}[\log \pi_\theta(a|s)]$$

is equivalent to minimizing the KL divergence:

$$\min_\theta \mathbb{E}_{s\sim p(s)}[D_{\mathrm{KL}}(\pi_g(\cdot|s)\|\pi_\theta(\cdot|s))].$$

Under assumption (iii) (Data Sufficiency), as the number of samples $N \to \infty$, the empirical distribution $\hat{p}(s, a)$ almost surely converges to the true distribution $p(s)\pi_g(a|s)$. Hence, the empirical optimization objective:

$$\frac{1}{N} \sum_{(s,a)\sim\hat{p}(s,a)} \log \pi_\theta(a|s)$$

almost surely converges to the true expectation:

$$\mathbb{E}_{s\sim p(s), a\sim\pi_g(a|s)}[\log \pi_\theta(a|s)].$$

Thus, under assumptions (i) (Sufficient Expressivity) and (ii) (Optimization Convergence), the optimization process applied to the empirical objective finds the global optimum that minimizes the expected KL divergence. Formally, we have:

$$\pi_{\theta^*} = \arg\min_\theta \mathbb{E}_{s\sim p(s)}[D_{\mathrm{KL}}(\pi_g(\cdot|s)\|\pi_\theta(\cdot|s))].$$

This completes the proof. □

## C.2   SFT Performance Bound in Terms of KL Divergence

In this subsection, we investigate the performance gap between the supervised fine-tuning (SFT) policy $\pi_{\mathrm{SFT}}$ and the fixed target policy $\pi_g$. Our goal is to establish that the difference in expected return between any policy $\pi$ and $\pi_g$ is controlled by their expected KL divergence. This provides a theoretical justification that SFT, which minimizes the KL divergence to $\pi_g$, achieves performance within $O(\sqrt{\kappa})$ of the teacher policy $\pi_g$.

Let the downstream metric $f(a \mid s)$ be bounded in $[0, R_{\max}]$, and define the performance of a policy $\pi$ as

$$J(\pi) := \mathbb{E}_{s\sim p(s)} \mathbb{E}_{a\sim\pi(\cdot|s)}[f(a \mid s)].$$

Then, we have the following theorem, which is a simplified version of the classic performance difference theorem (Schulman et al., 2015).

**Theorem 1** (Performance Difference Upper Bound)**.** *Let $\pi$ be any policy and $\pi_g$ be the data-generating (teacher) policy. Suppose the downstream metric is bounded by $0 \le f(a \mid s) \le R_{\max}$. Then the performance difference satisfies*

$$|J(\pi) - J(\pi_g)| \le R_{\max}\sqrt{\tfrac{1}{2} \mathbb{E}_{s\sim p(s)} \left[D_{\mathrm{KL}}\big(\pi_g(\cdot \mid s) \,\|\, \pi(\cdot \mid s)\big)\right]}.$$

*Proof.* Fix a state $s$. Define

$$P(a) = \pi_g(a \mid s), \quad Q(a) = \pi(a \mid s), \quad g_s(a) = \tfrac{f(a|s)}{R_{\max}} \in [0, 1].$$

The reward difference at $s$ can be written as

$$\Delta(s) = \mathbb{E}_{a\sim Q}[f(a \mid s)] - \mathbb{E}_{a\sim P}[f(a \mid s)] = R_{\max}\Big(\mathbb{E}_Q[g_s] - \mathbb{E}_P[g_s]\Big).$$

By the dual representation of total variation,

$$|\Delta(s)| \le R_{\max} \operatorname{TV}\big(\pi(\cdot \mid s), \pi_g(\cdot \mid s)\big).$$

Since the performance of a policy $\pi$ and $\pi_g$ is defined as

$$J(\pi) = \mathbb{E}_{s\sim p} \mathbb{E}_{a\sim Q}[f(a \mid s)], \qquad J(\pi_g) = \mathbb{E}_{s\sim p} \mathbb{E}_{a\sim P}[f(a \mid s)].$$

Then we can write

$$J(\pi) - J(\pi_g) = \mathbb{E}_{s\sim p} \Delta(s).$$

By applying the inequality $|\mathbb{E}[X]| \leq \mathbb{E}[|X|]$, we obtain

$$|J(\pi) - J(\pi_g)| \;=\; \left|\mathbb{E}_{s \sim p}[\Delta(s)]\right| \;\leq\; \mathbb{E}_{s \sim p}[|\Delta(s)|] \;\leq\; R_{\max}\, \mathbb{E}_{s \sim p}\big[\mathrm{TV}\big(\pi(\cdot \mid s),\, \pi_g(\cdot \mid s)\big)\big].$$

Next, by Pinsker's inequality, for each $s$ we have

$$\mathrm{TV}\big(\pi(\cdot \mid s), \pi_g(\cdot \mid s)\big) \leq \sqrt{\tfrac{1}{2}\, D_{\mathrm{KL}}\big(\pi_g(\cdot \mid s) \,\|\, \pi(\cdot \mid s)\big)}.$$

Averaging over $s$ and applying Jensen's inequality for the concave square root, we conclude

$$\mathbb{E}_s[\mathrm{TV}] \leq \sqrt{\tfrac{1}{2}\, \mathbb{E}_s[D_{\mathrm{KL}}(\pi_g(\cdot \mid s) \,\|\, \pi(\cdot \mid s))]}.$$

Combining these inequalities gives the desired bound.

Done. $\qquad\qquad\qquad\qquad\qquad\qquad\qquad\qquad\qquad\qquad\qquad\qquad\qquad\qquad\qquad\qquad$ □

**Implication for $\pi_{\mathrm{SFT}}$.** Since Theorem 1 established that $\pi_{\mathrm{SFT}}$ minimizes the expected KL divergence to $\pi_g$, the above result implies that the performance difference between $\pi_{\mathrm{SFT}}$ and $\pi_g$ is at most $O(\sqrt{\kappa})$, where

$$\kappa = \mathbb{E}_{s \sim p(s)}\left[D_{\mathrm{KL}}(\pi_g(\cdot \mid s) \,\|\, \pi_{\mathrm{SFT}}(\cdot \mid s))\right].$$

Thus, $\pi_{\mathrm{SFT}}$'s performance is limited by the teacher policy $\pi_g$.

### C.3 Dominance of RL Objective over SFT

We now compare the performance of the REC-R1 solution $\pi_{\mathrm{RL}}$ with that of the SFT solution $\pi_{\mathrm{SFT}}$. Recall that the performance of a policy $\pi$ is defined as

$$J(\pi) \;=\; \mathbb{E}_{s \sim p(s), a \sim \pi(\cdot \mid s)}[f(a \mid s)],$$

Then, we have the following theorem.

**Theorem 2** (Superiority of RL over SFT)**.** *Let $\pi_g$ denote the teacher policy, and let $\pi_{\mathrm{SFT}} \in \Pi$ be the supervised fine-tuning solution obtained by minimizing $\mathbb{E}_{s \sim p(s)}[D_{\mathrm{KL}}(\pi_g(\cdot \mid s) \| \pi(\cdot \mid s))]$ over the policy class $\Pi$. Let $\pi_{\mathrm{RL}} \in \arg\max_{\pi \in \Pi} J(\pi)$ denote the RL solution, i.e., the policy in $\Pi$ that maximizes the expected reward. Then*

$$J(\pi_{\mathrm{RL}}) \;\geq\; J(\pi_{\mathrm{SFT}}).$$

*Proof.* From Theorem 1, we know that the performance gap between $\pi_{\mathrm{SFT}}$ and the teacher policy $\pi_g$ satisfies

$$|J(\pi_{\mathrm{SFT}}) - J(\pi_g)| \;\leq\; \varepsilon_{\mathrm{SFT}},$$

where $\varepsilon_{\mathrm{SFT}} = R_{\max}\sqrt{\tfrac{1}{2}\kappa}$ and $\kappa = \mathbb{E}_{s \sim p(s)}[D_{\mathrm{KL}}(\pi_g(\cdot \mid s) \| \pi_{\mathrm{SFT}}(\cdot \mid s))]$.

Since $\pi_{\mathrm{RL}}$ is by definition the maximizer of $J(\pi)$ over $\Pi$, and $\pi_{\mathrm{SFT}} \in \Pi$, we directly have

$$J(\pi_{\mathrm{RL}}) \;\geq\; J(\pi_{\mathrm{SFT}}).$$

Done. $\qquad\qquad\qquad\qquad\qquad\qquad\qquad\qquad\qquad\qquad\qquad\qquad\qquad\qquad\qquad\qquad$ □

**Implication.** This theorem shows that reinforcement learning (REC-R1), by explicitly optimizing the downstream objective $J(\pi)$, is guaranteed to achieve performance no worse than SFT. In particular, while $\pi_{\mathrm{SFT}}$ is constrained to imitate $\pi_g$ (with performance at most $O(\sqrt{\kappa})$ close to the teacher), the RL policy $\pi_{\mathrm{RL}}$ can potentially surpass the teacher by exploiting exploration and reward optimization.

# D  Estimated Cost of SFT Data Generation and Rec-R1 Model Training

To assess the cost-effectiveness of our approach, we compare it against a supervised fine-tuning (SFT) baseline that relies on GPT-4o-generated instruction data. The SFT pipeline requires two stages: (1) generating 5,408 samples using GPT-4o, which costs approximately **$10.82** based on OpenAI's pricing ($2.50 per million input tokens and $10.00 per million output tokens), and (2) training the model on two NVIDIA A100 GPUs for **35 minutes** (2 epochs), which adds an additional **$4.78** (at an on-demand rate of $4.10/hour per A100 via AWS). The total cost for SFT amounts to approximately **$15.60**.

In contrast, REC-R1 requires no external data generation: it trains directly on synthetic data produced online by itself during learning. Remarkably, with only about **210 seconds** of training on the same two A100 GPUs (costing just **$0.48**), REC-R1 already matches and even surpasses the performance of the SFT-trained model. Moreover, performance continues to improve with further training. This comparison highlights the substantial cost-efficiency of REC-R1: it achieves superior performance at less than **1/30** of the SFT pipeline cost.

While our cost comparison is conducted on a small-scale experiment, the implications become more pronounced in real-world deployments. Supervised fine-tuning methods typically require generating millions of training examples and running long training cycles, leading to substantial costs in both data creation and computation. In contrast, REC-R1 eliminates the need for offline data generation and learns efficiently through online interaction, making it a significantly more cost-effective and practical solution for large-scale recommendation systems.

# E  Additional Experiment and Results Details

## E.1  Product Search

### E.1.1  Dataset Details

For the conventional setting, we use the ESCI dataset processed by Hou et al. (2024a), a benchmark derived from Amazon product search logs. Following their preprocessing protocol, we focus on four representative product categories: *Video Games*, *Baby Products*, *Office Products*, and *Sports and Outdoors*. We construct category-specific splits with 4,510 training examples, 898 validation examples, and 798 test examples. For all ESCI experiments, we use the same item pool as Hou et al. (2024a), which contains 1,367,729 product listings. Each category uses the full item pool for retrieval.

For the complex setting, we use the Amazon-C4 dataset introduced in Hou et al. (2024a), which contains complex natural language product queries. Since the released dataset provides only category-labeled test queries, we treat the four domains used in ESCI as our test set, and use queries from all other domains as the training and validation data. This results in 18,126 training examples, 2,722 validation examples, and 1,722 test examples. The corresponding item pool consists of 1,058,417 products, identical to that used by Hou et al. (2024a). As with ESCI, each domain-specific split uses the full item pool for retrieval. This cross-domain setup allows us to evaluate the generalization capability of REC-R1 when applied to unseen product categories in a more realistic, open-ended retrieval scenario.

### E.1.2  Implementation Details

We implement REC-R1 using the VeRL library[1], and run all experiments on two NVIDIA A100 80GB GPUs.

**Retriever Setup.** We support both sparse and dense retrievers in our framework. For sparse retrieval, we use Pyserini (Lin et al., 2021) with Lucene's BM25 implementation. Following standard practice, we set the BM25 hyperparameters as $k_1 = 1.2$ and $b = 0.75$. For dense retrieval, we build HNSW-based FAISS (Johnson et al., 2019) indices. The dense embeddings are first L2-normalized to enable cosine similarity search. We use IndexHNSWFlat with $M = 32$ and efConstruction=200 to balance search accuracy and indexing speed.

---

[1]https://github.com/volcengine/verl

Table 5: Prompt used in Rec-R1 for product search tasks with BM25, where the LLM generates structured query terms based on a user query.

---

**Prompt Template for Rec-R1 + BM25 (Product Search)**

---

```
<|im_start|>system
You are a helpful AI assistant.  You first think about the reasoning process in the
mind and then provide the user with the answer.
<|im_end|>
<|im_start|>user
You are an expert in query generation.  Given a query, your task is to create query
terms to retrieve the most relevant products, ensuring they best meet customer needs.

Below is the query:
``` {user_query} ```

Show your work in <think>\think> tags.  Your final response must be in JSON format
within <answer>\answer> tags.  The generated query should use Boolean operators (AND,
OR) to structure your query logically.  For example,
<answer>
{ "query":  xxx }
</answer>.
<|im_end|>
<|im_start|>assistant
Let me solve this step by step.
<think>
```

---

**Training Configuration.** We use Group Relative Policy Optimization (GRPO) as our reinforcement learning algorithm, following the implementation in VeRL. The language model is initialized from Qwen-2.5-3B-Instruct, and optimized with KL-regularized policy gradients. To control policy divergence, we apply a low-variance KL loss with coefficient 0.001.

Each prompt is used to generate 12 sampled responses using top-$p$ sampling ($p = 0.95$) and temperature 0.6. The rollout engine uses vLLM with memory budget capped at 30% GPU utilization. Training is run for 5 epochs with a learning rate of 1e−6, global mini-batch size of 256, and micro-batch size of 2. We also enable gradient checkpointing and use Fully Sharded Data Parallelism (FSDP) with parameter and gradient offloading for memory efficiency.

We use NDCG@1000 as the reward during training (instead of NDCG@100 at evaluation time) to reduce reward sparsity and stabilize learning. All other parameters follow VeRL defaults unless otherwise specified.

The prompts for product search can be found in Table 5 for BM25 and Table 6 and 7 for Dense retrievers.

### E.1.3 Additional Analysis: The Role of Prompt Design and Exploration in RL

In this section, we investigate how different prompt strategies affect retrieval performance when paired with dense retrievers—particularly BLAIR, which is pretrained using user reviews and item metadata through contrastive learning. Our hypothesis is that aligning the generation style of the query rewriting process with the pretraining distribution of the retriever could lead to better synergy and downstream performance.

As shown in Table 8, using generic prompts (Table 6) to rewrite queries into natural language variations yields limited or even negative impact across all models, including GPT-4o and Qwen-2.5-3B-Instruct. Moreover, initializing Rec-R1 with such rewriting behavior also results in modest performance.

To address this, we experimented with a more targeted prompt strategy: instructing the model to convert the input query into a user-style review (Table 7), which better mirrors the training data format used by BLAIR. Interestingly, without reinforcement learning, this "review-style rewriting" strategy alone often hurts performance. However, when training Rec-R1 under this revised prompting strategy, we observe that

Table 6: Prompt used in REC-R1 for product search tasks with dense retrievers, where the LLM expands the original query with semantically relevant information to improve retrieval.

| **Prompt Template for Rec-R1 + Dense Retriever (Product Search)** |
|---|

```
<|im_start|>system
You are a helpful AI assistant.  You first think about the reasoning process in the
mind and then provide the user with the answer.
<|im_end|>
<|im_start|>user
You are an expert in generating queries for dense retrieval.  Given a customer
query, your task is to retain the original query while expanding it with additional
semantically relevant information, retrieve the most relevant products, ensuring
they best meet customer needs.  If no useful expansion is needed, return the original
query as is.

Below is the query:
``` {user_query} ```

Show your work in <think>\think> tags.  Your final response must be in JSON format
within <answer>\answer> tags.  For example,
<answer>
{ "query":  xxx }
</answer>.
<|im_end|>
<|im_start|>assistant
Let me solve this step by step.
<think>
```

the model gradually learns to generate review-style queries that significantly enhance retrieval performance. REC-R1 not only recovers from the initially degraded performance but also surpasses all baselines across all four domains, achieving new state-of-the-art results with both BLAIR-BASE and BLAIR-LARGE.

This experiment also sheds light on the importance of guided exploration in reinforcement learning, especially in language generation tasks with extremely large action spaces. In our setting, the LLM selects an action at each token position, and the final output is a long sequence—meaning the search space is exponentially large. Without meaningful initial guidance, for those very hard problems, early exploration can easily fall into suboptimal regions, from which recovery is difficult due to sparse or misleading reward signals.

### E.1.4 Comparison with Other Fine-Tuning Strategies: Rejection Sampling and DPO

To better understand the strengths of our method, we compared REC-R1 against two widely used fine-tuning strategies—Rejection Sampling Fine-Tuning and Direct Preference Optimization (DPO)—within the ESCI + BM25 setting. Table 9 summarizes the NDCG performance across four product domains.

*Rejection Sampling Fine-Tuning* involved filtering training examples where outputs from top-performing LLMs (GPT-4o, Claude-3.5-Sonnet, Claude-3-Haiku) achieved an NDCG greater than 0.5. These examples were then used to fine-tune the Qwen-2.5-3B-Instruct model.

*DPO* was trained by selecting, for each training instance, the best and worst candidate queries (by NDCG) from the same LLM pool. The model was fine-tuned for one epoch using these pairs. However, the results were substantially lower across all domains.

From the results, we can observe a clear performance gap between REC-R1 and the two baseline fine-tuning methods. While DPO provides a theoretically grounded approach to preference learning, its reliance on static preference pairs—without any feedback from the downstream task environment—limits its generalization

Table 7: Prompt used in REC-R1 for Amazon-C4 dense retrieval with BLAIR, where the LLM rewrites queries into review-style texts aligned with BLAIR's pretraining objective.

---

**Prompt Template for Rec-R1 + Dense Retriever (Amazon-C4, Review-Style Rewriting)**

---

```
<|im_start|>system
You are a helpful AI assistant.  You first think about the reasoning process in the
mind and then provide the user with the answer.
<|im_end|>
<|im_start|>user
You are an expert in query rewriting for dense retrieval systems.  Rewrite the
following product search query as if you are a real customer writing a natural,
authentic review after using the product.  Maintain the meaning and details of
the original query, but shift the tone to be more casual, emotional, and based on
personal experience.  Include specific comments about product performance that match
the query's intent.

# Below is the product search query:
# ```{user_query} ```

Show your work in <think> </think> tags.  Your final response must be in JSON format
within <answer> </answer> tags.  For example,
<answer>
{ "query":  xxx }
</answer>.
<|im_end|>
<|im_start|>assistant
Let me solve this step by step.
<think>
```

---

ability. The poor performance of DPO across all domains suggests that, in complex ranking scenarios, such offline preference supervision may be insufficient and prone to overfitting.

Rejection sampling achieves moderately better results. However, this approach still lacks a mechanism to iteratively refine the policy based on task-specific feedback. In contrast, REC-R1 benefits from a closed-loop optimization process, where model updates are continually guided by performance within the recommendation environment.

### E.1.5 Applicability to Alternative Backbone Models

To evaluate the generalizability of REC-R1, we conducted additional experiments using a variety of backbone models beyond Qwen-2.5-3B. Specifically, we tested REC-R1 on smaller models such as Qwen-2.5-0.5B and Qwen-2.5-1.5B, as well as on a different model family—LLaMA-3.2-3B. All experiments were performed under the same ESCI + BM25 setting, with results presented in Table 10.

The results show a consistent and significant performance boost when applying REC-R1 across all backbone sizes and architectures. For instance, REC-R1-0.5B achieves an average improvement of over 13 points in NDCG compared to the base Qwen-2.5-0.5B model, outperforming even some larger models. Similarly, REC-R1 applied to LLaMA-3.2-3B yields results comparable to or better than its Qwen-based counterpart, despite architectural differences.

### E.1.6 Comparison with Modern Strong Query Rewriting Baselines

We further compared REC-R1 with several recent and strong LLM-assisted query rewriting and expansion baselines on both the ESCI dataset and the Amazon-C4 corpus with BM25 retrieval. The compared baselines

Table 8: Performance comparison between general query rewriting and review-style query rewriting under BLAIR-Base and BLAIR-Large. We observe that while prompting alone offers marginal or negative gains, REC-R1 achieves significant improvements—especially when aligned with the inductive biases of dense retrievers (e.g., BLAIR pre-trained on review-style text).

| Model | Video Games | Baby Products | Office Products | Sports and Outdoors |
|---|---|---|---|---|
| **Base retriever** | | | | |
| BLAIR$_{\text{BASE}}$ | 19.14 | 19.53 | 17.43 | 20.02 |
| BLAIR$_{\text{LARGE}}$ | **24.86** | **22.44** | 18.92 | **24.54** |
| **Using general query rewriting prompts** | | | | |
| GPT-4o$_{+\text{BLAIR-BASE}}$ | 21.12 | 19.54 | 17.22 | 22.68 |
| GPT-4o$_{+\text{BLAIR-LARGE}}$ | **23.40** | 21.00 | 18.69 | **25.74** |
| Qwen-2.5-3B-Instruct$_{+\text{BLAIR-BASE}}$ | 15.82 | 18.07 | 14.34 | 16.96 |
| Qwen-2.5-3B-Instruct$_{+\text{BLAIR-LARGE}}$ | 18.20 | 19.19 | 15.37 | 18.94 |
| REC-R1-3B$_{+\text{BLAIR-BASE}}$ | 19.65 | 20.85 | 18.91 | 22.29 |
| REC-R1-3B$_{+\text{BLAIR-LARGE}}$ | 19.26 | 21.63 | 18.93 | 21.64 |
| **Models convert queries into reviews** | | | | |
| GPT-4o$_{+\text{BLAIR-BASE}}$ | 20.61 | 19.35 | 17.63 | 21.74 |
| GPT-4o$_{+\text{BLAIR-LARGE}}$ | 16.59 | 15.14 | 15.05 | 16.76 |
| Qwen-2.5-3B-Instruct$_{+\text{BLAIR-BASE}}$ | 19.40 | 16.73 | 15.59 | 18.50 |
| Qwen-2.5-3B-Instruct$_{+\text{BLAIR-LARGE}}$ | 22.06 | 18.31 | 16.51 | 20.65 |
| REC-R1-3B$_{+\text{BLAIR-BASE}}$ | 21.69 | **25.62** | **22.17** | 24.22 |
| REC-R1-3B$_{+\text{BLAIR-LARGE}}$ | 26.51 | 27.04 | 23.10 | 27.40 |

Table 9: Comparison of NDCG performance across product domains using different fine-tuning strategies under the ESCI + BM25 setup.

| Model | Video Games | Baby Products | Office Products | Sports and Outdoors |
|---|---|---|---|---|
| Qwen-2.5-3B-Rej-Sample | 22.92 | 20.67 | 27.01 | 24.89 |
| Qwen-2.5-3B-DPO | 14.53 | 12.57 | 13.52 | 13.32 |
| REC-R1-3B | **33.89** | **29.27** | **34.61** | **31.92** |

include DocT5Query (Nogueira et al., 2019), RetPO (Yoon et al., 2025), LLM4CS (Mao et al., 2023), and AdaQR (Zhang et al., 2024). Results are summarized in Tables 11 and 12.

On ESCI, REC-R1 achieves substantial gains over all baselines across the four domains. On Amazon-C4, the performance gap is even more striking. REC-R1 surpasses all competing approaches by large margins. Overall, these results clearly demonstrate that REC-R1 consistently outperforms modern strong LLM-assisted query rewriting baselines.

### E.2 Sequential Recommendation

### E.2.1 Dataset Details

We conduct our sequential recommendation experiments on the Amazon Beauty dataset curated by Hou et al. (2024a). Following their protocol, we split the data chronologically based on absolute timestamps into training, validation, and test sets. The final splits include 96,778 training samples, 3,538 validation samples, and 1,538 test samples—comprising 1,000 inductive and 538 transductive test cases. All experiments use the Amazon Beauty item pool containing 43,982 unique products, consistent with the setting in Hou et al. (2024a).

### E.2.2 LLM Input Construction for Sequential Recommendation

To adapt the sequential recommendation task for use with generative LLMs, we convert each user's interaction history into a natural language format. Specifically, for each historical item, we concatenate the titles

Table 10: NDCG performance of REC-R1 across different backbone models under the ESCI + BM25 setting.

| Model | Video Games | Baby Products | Office Products | Sports and Outdoors |
|---|---|---|---|---|
| Qwen-2.5-0.5B | 10.88 | 12.93 | 20.21 | 17.09 |
| REC-R1-0.5B | 27.18 | 25.14 | 34.80 | 30.70 |
| Qwen-2.5-1.5B | 15.14 | 14.09 | 21.77 | 15.84 |
| REC-R1-1.5B | 31.68 | 26.46 | 32.92 | 30.87 |
| LLaMA-3.2-3B | 19.16 | 16.87 | 19.35 | 16.90 |
| REC-R1-3B (LLaMA) | 32.41 | 29.40 | 34.26 | 31.39 |

Table 11: Comparison with modern query rewriting baselines on the ESCI + BM25 setting. We report NDCG@100 across four domains. The best performance is denoted in **bold**.

| Model | Video Games | Baby Products | Office Products | Sports and Outdoors |
|---|---|---|---|---|
| DocT5Query | 15.02 | 14.37 | 17.19 | 16.83 |
| RetPO | 23.43 | 17.88 | 19.00 | 20.80 |
| LLM4CS | 23.29 | 24.02 | 28.06 | 28.00 |
| AdaQR | 23.53 | 20.59 | 29.02 | 26.36 |
| REC-R1 | **33.89** | **29.27** | **34.61** | **31.92** |

using newline characters (\n) as separators. To ensure compatibility with the LLM's input length limit (set to 512 tokens in our experiments), we retain only the latest 10 items from the history list. The resulting text sequence serves as the context for generation.

This processed history text is then formatted into a prompt for the LLM to generate a guess of the next item the user might want. An example of this prompt format is shown in Table 13.

### E.2.3 Implementation Details

The training setup for the sequential recommendation task largely mirrors that of product search, including the use of the VeRL library and two NVIDIA A100 80GB GPUs. One notable difference lies in the input length configuration: we set max prompt length to 512 (instead of 256), since each input includes a full user history list composed of multiple product titles, which tends to be significantly longer than single-turn queries in product search. All other hyperparameters remain unchanged.

### E.2.4 Additional Analysis: Why Rec-R1 Performs Better in the Inductive Setting?

Notably, we find that REC-R1 achieves stronger performance in the inductive setting compared to the transductive one—despite the latter having more item overlap with training data. This may seem counterintuitive at first, but we believe the reason lies in the nature of our framework and the task formulation.

In the transductive setting, many test items have already appeared in the training data. Traditional models can exploit this overlap through direct memorization or overfitting to item co-occurrence patterns. However, in REC-R1, the LLM is trained to infer the next item via natural language generation, which requires capturing underlying intent or semantics. When the task itself lacks a strong logical mapping from history to future items—as is often the case in sequential recommendation—language-based reasoning becomes less effective and may even introduce noise.

In contrast, the inductive setting removes such memorization shortcuts, as the target items are completely unseen during training. This forces the model to rely on more transferable semantic patterns, which better aligns with REC-R1 's learning mechanism. The LLM is incentivized to produce generalized, meaningful descriptions that reflect what kind of item could come next—rather than relying on item identity. As a result, the inductive setting provides a clearer signal for reward-driven optimization, enabling REC-R1 to shine where conventional models struggle.

Table 12: Comparison with modern query rewriting baselines on Amazon-C4 + BM25. We report NDCG@100 across four domains. The best performance is denoted in **bold**.

| Model | Video Games | Baby Products | Office Products | Sports and Outdoors |
|-------|-------------|---------------|-----------------|---------------------|
| DocT5Query | 3.90 | 5.81 | 4.93 | 6.43 |
| RetPO | 5.20 | 1.40 | 1.64 | 2.54 |
| LLM4CS | 13.05 | 9.97 | 11.51 | 11.05 |
| AdaQR | 12.98 | 9.15 | 10.47 | 10.41 |
| REC-R1 | **26.51** | **27.04** | **23.10** | **27.40** |

Table 13: Prompt format used for LLM-based generation in the sequential recommendation task. The input includes the user's purchase history and instructs the model to output structured query terms for the next likely purchase.

**Prompt Template Used for LLM Input in Sequential Recommendation**

```
<|im_start|>system
You are a helpful AI assistant.  You first think about the reasoning process in the
mind and then provide the user with the answer.
<|im_end|>
<|im_start|>user
You are an intelligent shopping assistant that helps predict what users may want to
purchase next.  Below is a list of items a user has purchased recently.  Your task
is to infer one or multiple kinds of products they may want to buy next, and generate
relevant query terms that can be used to search for these potential products.

Below is the user purchase history:
``` {purchase_history} ```

Show your work in <think>\think> tags.  Your final response must be in JSON format
within <answer>\answer> tags.  The generated query should use Boolean operators (AND,
OR) to structure your query logically.  For example,
<answer>
{ "query":  xxx }
</answer>.
<|im_end|>
<|im_start|>assistant
Let me solve this step by step.
<think>
```

### E.3 Product Re-ranking

### E.3.1 Dataset Details

We build the re-ranking dataset starting from the ESCI corpus curated by Hou et al. (2024a). For each split (train/validation/test), we first obtain initial candidate lists using the Rec-R1-Retriever trained in the Product Search task (Section 3.1), combined with BM25 retrieval. We select the top-16 items as the candidates. To ensure data quality, we filter out samples with zero retrieval performance, i.e., those for which the upstream retriever achieves NDCG@20 = 0. After filtering, each data point consists of the user query, the candidate list (input to the re-ranker), and the corresponding ground-truth relevant items. We finally obtain 2,166, 521, and 465 samples for the train, validation, and test splits, respectively.

Table 14: Prompt used for the product re-ranking task.

**Prompt Template for Rec-R1 + Product Re-ranking**

```
<|im_start|>system
You are a helpful AI assistant.  You first think about the reasoning process in the
mind and then provide the user with the answer.
<|im_end|>
<|im_start|>user
You are an expert in product reranking.  Given a customer query and a list of
candidate products, your task is to rerank the products so that the most relevant
items to the query are placed at the top.  Consider semantic meaning, product
attributes, and customer intent.

Below are the candidate products (item_id and metadata):
# ```{candidates} ```

Below is the query:
# ```{user_query} ```

Show your work in <think> </think> tags.  Your final response must be in JSON format
within <answer> </answer> tags.  The output JSON should contain the reranked list
of item_ids in order of relevance.  The length of the reranked list should match the
number of input candidates.  For example,
<answer>
{ "reranked_items":  ["Item_4", "Item_8", "Item_x", ..., all candidate item_ids in
order] }
</answer>
<|im_end|>
<|im_start|>assistant
Let me solve this step by step.
<think>
```

### E.3.2 Implementation Details

The overall training and evaluation setup follows the same configuration as in the product search and sequential recommendation tasks. We highlight the key differences below: (1) the LLM prompts are specifically designed for the re-ranking scenario, as shown in Table 14; and (2) the input/output length limits are adjusted, with max prompt length set to 3000 and max response length set to 1024.

### E.4 Evaluation of Generalization and Forgetting

### E.4.1 Implementation Details

To assess whether REC-R1 preserves the general-purpose capabilities of the underlying LLM while achieving strong task-specific performance, we evaluate all models across a suite of generalization benchmarks. Specifically, we consider six datasets spanning different task types and reasoning skills:

- **ESCI (NDCG@100)** – Product search recommendation, serving as the target task of optimization.

- **MMLU (Accuracy)** – A factual knowledge benchmark covering multiple-choice questions across various domains (Hendrycks et al., 2020).

- **IFEval (Strict Accuracy)** – A benchmark designed to evaluate instruction-following and alignment with user intent (Zhou et al., 2023).

- **GSM8K (5-shot, EM)** – Math reasoning with elementary school word problems in a few-shot setting, measured by exact match (Cobbe et al., 2021).

- **MBPP (3-shot, pass@1)** – A coding benchmark consisting of short Python problems, evaluated using pass@1 (Austin et al., 2021).

- **HumanEval (0-shot, pass@1)** – A high-quality Python programming test measuring zero-shot code generation performance (Chen et al., 2021).

All evaluations are conducted using the `lm-evaluation-harness` library (Gao et al., 2024) to ensure consistency and reproducibility. For ESCI, we directly compute NDCG@100 based on model-generated rewritings. For all other datasets, we use the official protocols defined in `lmeval`.

### E.4.2   Additional Analysis: Impact of Reasoning and JSON Format in SFT

To better understand the effects of prompt format on supervised fine-tuning (SFT), we explore four SFT variants that differ in whether the training data includes intermediate reasoning steps and whether the answers are wrapped in structured JSON format. We use GPT-4o-generated data on the ESCI product search task for training all four SFT models. Table 15 shows results on ESCI, and Table 16 evaluates generalization to broader benchmarks.

On the task-specific ESCI dataset, all SFT variants outperform the base model (Qwen-2.5-3B-Instruct), demonstrating the effectiveness of supervised fine-tuning on task-specific data. However, all variants fall short compared to REC-R1, which uses the same data but trains via reinforcement learning. This highlights the advantage of reward-driven learning in aligning with downstream task metrics.

We then assess the general-purpose capabilities of these models. On MMLU, a knowledge-intensive benchmark, all SFT variants retain performance close to the original model (within 2 points), suggesting factual knowledge is preserved. In contrast, IFEval results reveal **catastrophic forgetting across the board—all SFT variants** suffer 20–30 point drops in instruction-following accuracy, regardless of format. This underscores the risk of overfitting in SFT, where tuning on narrow task data compromises broader generalization.

An interesting observation arises on GSM8K: the variant with JSON formatting but no reasoning shows improved performance over the base model (+4.7). We hypothesize that the strict output format (JSON) acts as a "shield," isolating the fine-tuning effects from interfering with the model's native reasoning process. In contrast, the reasoning-heavy variants modify the generative behavior more substantially, harming out-of-domain reasoning.

On the coding benchmarks (MBPP, HumanEval), all four SFT variants exhibit comparable or slightly improved performance relative to the original model—regardless of whether the training outputs used JSON format. This suggests that coding ability is relatively robust to task-specific SFT, and may even benefit from it. One possible explanation is that the ESCI task, although unrelated to coding, implicitly encourages structured generation and logical formatting (e.g., conditionally constructed queries or Boolean expressions), which aligns well with the formal nature of code generation.

In contrast, REC-R1 avoids these trade-offs altogether. **Rec-R1 improves task-specific performance while maintaining general capabilities across reasoning, knowledge, and code generation.** These results provide further evidence that REC-R1 is a more stable and generalizable learning framework than conventional SFT.

### E.5   Evaluations on Standard IR Benchmarks

To further examine the generality of our framework, we conducted evaluations on several widely used information retrieval (IR) benchmarks, including MS MARCO (Nguyen et al., 2016), two representative datasets from BEIR (Thakur et al., 2021) (NFCorpus and FEVER), as well as the TREC Deep Learning tracks DL'19 and DL'20 (Dai et al., 2024). Results are summarized in Table 17. Across all benchmarks, REC-R1 consistently outperforms the baselines. These results further validate the effectiveness and robustness of our framework under standard IR benchmarks, even though retrieval is not the primary focus of this paper.

Table 15: Performance on ESCI datasets across different domains. Note: The REC-R1 results here may differ slightly from Table 1 due to using the checkpoint at step 1400 instead of step 1390. This minor difference has negligible impact on performance. We report the performance of models under different training steps because evaluation on validation and test set is done every 10 steps and models are saved every 100 steps. We use the 1400-step checkpoint for consistency in follow-up experiments.

| Model | Reasoning | JSON | ESCI (Games) | ESCI (Baby) | ESCI (Office) | ESCI (Sports) |
|---|---|---|---|---|---|---|
| Qwen-2.5-3B-Instruct | - | - | 19.63 | 16.03 | 19.96 | 21.36 |
| Qwen-2.5-3B-Instruct (SFT) | ✓ | ✓ | 25.70 | 19.66 | 27.34 | 24.82 |
| Qwen-2.5-3B-Instruct (SFT) | ✗ | ✓ | 26.87 | 22.83 | 26.10 | 26.42 |
| Qwen-2.5-3B-Instruct (SFT) | ✗ | ✗ | 25.08 | 21.50 | 28.75 | 25.18 |
| Qwen-2.5-3B-Instruct (SFT) | ✓ | ✗ | 23.31 | 20.77 | 24.34 | 24.78 |
| REC-R1-3B (1400 steps) | - | - | 32.92 | 29.62 | 35.05 | 31.21 |

Table 16: Performance on general-purpose reasoning and coding benchmarks. Color-coded deltas show change from baseline Qwen-2.5-3B-Instruct.

| Model | Reasoning | JSON | MMLU | IFEval | GSM8K | MBPP | HumanEval |
|---|---|---|---|---|---|---|---|
| Qwen-2.5-3B-Instruct | - | - | 65.4 | 58.2 | 63.4 | 53.6 | 46.3 |
| Qwen-2.5-3B-Instruct (SFT) | ✓ | ✓ | 63.7 (-1.7) | 31.4 (-26.8) | 35.7 (-27.7) | 54.8 (+1.2) | 53.6 (+7.3) |
| Qwen-2.5-3B-Instruct (SFT) | ✗ | ✓ | 64.1 (-1.3) | 30.5 (-27.7) | 68.1 (+4.7) | 50.4 (-3.2) | 57.3 (+11.0) |
| Qwen-2.5-3B-Instruct (SFT) | ✗ | ✗ | 64.3 (-1.1) | 34.4 (-23.8) | 37.3 (-26.1) | 52.2 (-1.4) | 53.6 (+7.3) |
| Qwen-2.5-3B-Instruct (SFT) | ✓ | ✗ | 63.6 (-1.8) | 35.0 (-23.2) | 50.2 (-13.2) | 55.4 (+1.8) | 52.4 (+6.1) |
| REC-R1-3B (1400 steps) | - | - | 65.3 (-0.1) | 60.1 (+1.9) | 69.1 (+5.7) | 54.4 (+0.8) | 46.3 (+0.0) |

## E.6 Comparison with RL-for-Search Approaches

Recent studies have also applied reinforcement learning with verifiable reward (RLVR) for search. We highlight two representative approaches—Search-R1 (Jin et al., 2025) and ConvSearch-R1 (Zhu et al., 2025)—and compare them against our proposed Rec-R1.

**Search-R1.** As described in Section 3.4 of their paper, Search-R1 relies solely on a single exact-matching reward. In contrast, Rec-R1 directly optimizes downstream recommendation metrics (e.g., Recall, NDCG) as the reward function. This distinction is crucial: due to the nature of RLVR optimization, Search-R1 cannot explicitly constrain the quality of the intermediate retrieval process, while Rec-R1 is directly aligned with the final RecSys evaluation objectives. As shown in Tables 18 and 19, Rec-R1 consistently and significantly outperforms Search-R1 across all domains in both ESCI and Amazon-C4 datasets.

**ConvSearch-R1.** ConvSearch-R1 differs in its reward design: instead of directly optimizing evaluation metrics, it introduces a piecewise-defined reward function (Eq. (1) in their paper). Their motivation is to mitigate reward sparsity, arguing that optimizing NDCG@3 directly would result in too few positive signals. However, we argue that such sparsity is only a problem if one insists on optimizing at a very small cutoff (e.g., $k = 3$). In fact, optimizing at larger $k$ implicitly improves smaller-$k$ NDCG as well. Moreover, ConvSearch-R1's reward formulation does not directly align with evaluation metrics. Empirically, Rec-R1 consistently outperforms ConvSearch-R1 on both ESCI and Amazon-C4 (Tables 18 and 19).

In summary, Rec-R1 distinguishes itself by (i) reward design that is directly tied to RecSys evaluation metrics, and (ii) consistently stronger empirical performance on LLM4Rec tasks compared to recent RL-for-search baselines.

## E.7 Rec-R1 under Multi-Objective Reward Optimization

The reward structure in REC-R1 is designed to be general and flexible. Formally, the reward is defined as a function $f(a|s)$, where $s$ denotes the input state and $a$ the model output. This formulation naturally

Table 17: **Performance on standard IR benchmarks.** We report MRR@10 for MS MARCO and NDCG@10 for BEIR and TREC DL datasets. The best performance is denoted in **bold**.

| Model | MS MARCO | NFCorpus | FEVER | DL'19 | DL'20 |
|---|---|---|---|---|---|
| BM25 | 44.80 | 14.67 | 44.25 | 42.31 | 47.80 |
| Qwen2.5-3B-Instruct | 23.00 | 29.23 | 46.68 | 43.56 | 52.24 |
| GPT-4o | 39.43 | 30.67 | 53.94 | 59.87 | 54.72 |
| **Rec-R1** | **53.13** | **34.00** | **66.38** | **63.25** | **59.71** |

Table 18: **Comparison with RL-for-search approaches on ESCI + BM25 (NDCG@100).** Rec-R1 consistently outperforms both Search-R1 and ConvSearch-R1 across all domains.

| Model | Video Games | Baby Products | Office Products | Sports and Outdoors |
|---|---|---|---|---|
| Search-R1 | 18.03 | 17.38 | 22.67 | 19.62 |
| ConvSearch-R1 | 23.67 | 22.35 | 27.89 | 25.64 |
| **Rec-R1** | **33.89** | **29.27** | **34.61** | **31.92** |

supports multi-objective optimization: $f(a|s)$ can be extended to incorporate multiple reward signals via linear combination, such that the optimization encourages trajectories with higher aggregated rewards.

To further validate the flexibility of our framework, we conducted additional experiments to examine whether Rec-R1 can be stably trained under a multi-reward setting. Specifically, we combined two types of rewards: (i) a *downstream metric reward* (e.g., Recall or NDCG), which measures recommendation quality; (ii) a *format reward*, which ensures well-structured and valid outputs; (iii) *Category Consistency Reward*, which measures the proportion of retrieved items belonging to the same category as the ground-truth items given a query. The results are shown in Table 20. Rec-R1 continues to perform strongly across all domains, substantially outperforming baseline LLMs such as Qwen2.5-3B-Instruct and GPT-4o, while maintaining training stability.

These findings indicate that Rec-R1 is not limited to single-metric optimization but can be readily extended to multi-objective reward structures, offering a flexible framework for incorporating diverse recommendation signals.

## F  Discussion and Future Directions

In this section, we reflect on several insights and limitations emerging from our experiments, and highlight future directions for building stronger recommendation-oriented LLMs.

**LLMs Can Learn to Recommend Without Access to the Item Space.** In our product search experiments, REC-R1 operates without any access to the downstream item catalog—it only receives the user query $s$ and generates a rewritten query $a$, without knowing which products exist in the recommender's database. Despite this apparent limitation, REC-R1 consistently delivers strong performance across domains. This aligns surprisingly well with human behavior: when people search for products, they rarely know the exact contents of a platform's inventory. Instead, they refine their queries iteratively based on vague goals and system feedback. REC-R1, trained in a closed loop with the recommender, learns this refinement process efficiently via reinforcement learning. This result highlights the potential of LLMs to simulate user-side reasoning, making them powerful agents for optimizing recommender interaction.

**Toward Better Integration with Sequential Recommendation.** Our sequential recommendation setup frames the LLM as a next-item predictor: it receives a history of product titles and generates a guess of the next likely item in natural language, which is then fed to a retriever. While this approach works well in the inductive setting—where test items are unseen—it underperforms traditional models in the transductive case. We have previously explain it in §E.2.4. A promising direction is to use LLMs not as generation agents, but as feature augmentation modules: the LLM could enrich each item in the user history with additional

Table 19: **Comparison with RL-for-search approaches on Amazon-C4 + BM25 (NDCG@100).** Rec-R1 achieves significant improvements across domains.

| Model | Video Games | Baby Products | Office Products | Sports and Outdoors |
|---|---|---|---|---|
| Search-R1 | 12.06 | 9.12 | 10.21 | 9.96 |
| ConvSearch-R1 | 13.57 | 13.08 | 13.62 | 13.34 |
| **Rec-R1** | **26.51** | **27.04** | **23.10** | **27.40** |

Table 20: **Performance under multi-objective reward setting on ESCI + BM25 (NDCG@100).** Rec-R1 combines format reward, NDCG, and category consistency reward.

| Model | Video Games | Baby Products | Office Products | Sports and Outdoors |
|---|---|---|---|---|
| Qwen2.5-3B-Instruct | 19.63 | 16.03 | 19.96 | 21.36 |
| GPT-4o | 26.06 | 23.05 | 27.98 | 27.38 |
| **Rec-R1 (multi-reward)** | **33.06** | **29.39** | **34.22** | **32.27** |

descriptive or contextual information. These enriched sequences could then be encoded using text encoders like BLAIR and passed into standard SRec models. This hybrid approach could combine LLMs' semantic understanding with the modeling power of sequential architectures for the transductive setting.

**Initialized LLM's capabilities Matter.** Our findings underscore the importance of the base LLM's capabilities when applying reinforcement learning to complex decision-making tasks. Like traditional RL pipelines that rely on imitation learning to bootstrap strong initial behaviors—e.g., in high-stakes environments like Go and MOBA games (Silver et al., 2016; Vinyals et al., 2019)—Rec-R1 also benefits significantly from a well-initialized model. In our experiments, Qwen-2.5-3B-Instruct provides a strong general-purpose foundation. However, we also observe that this general strength does not guarantee effectiveness on every domain-specific task. For instance, in sequential recommendation, the base model lacks any prior experience in predicting the next item from user histories—resulting in a weak starting point for RL-based optimization. This raises a compelling direction: could domain-specific pretraining or instruction tuning (e.g., training LLMs to imitate existing sequential recommender outputs) better equip models to serve as Rec-R1 agents? Combining domain-aware LLMs with Rec-R1 could unlock more powerful and semantically aligned generation strategies, especially for tasks like sequential recommendation. We leave it as a future work.

**Leveraging RecSys Feedback: From Static Logs to Live Interaction.** A core advantage of Rec-R1 lies in its ability to leverage feedback signals from recommendation systems. In our experiments, these signals are derived from historical interaction logs that are commonly available in large-scale recommender platforms. While the current feedback is log-based, this setup aligns well with real-world deployment, where user interactions are continuously collected in vast quantities. In practice, maintaining an up-to-date stream of logs allows Rec-R1 to stay aligned with evolving user preferences and content trends. Moreover, Rec-R1 is fully compatible with real-time feedback: it can be trained via online interactions with a live recommendation engine, where the LLM receives immediate performance signals (e.g., engagement rates or conversions). This makes Rec-R1 a flexible framework capable of serving as a foundation for LLM-based recommendation systems that evolve with real-world usage.

# G Case Study

To better understand the behavior and effectiveness of Rec-R1, we present qualitative case studies from the **product search** task (ESCI dataset) and the **sequential recommendation** task (Amazon Beauty dataset). These cases offer insights into how Rec-R1 generates more effective textual inputs than prompting-based methods.

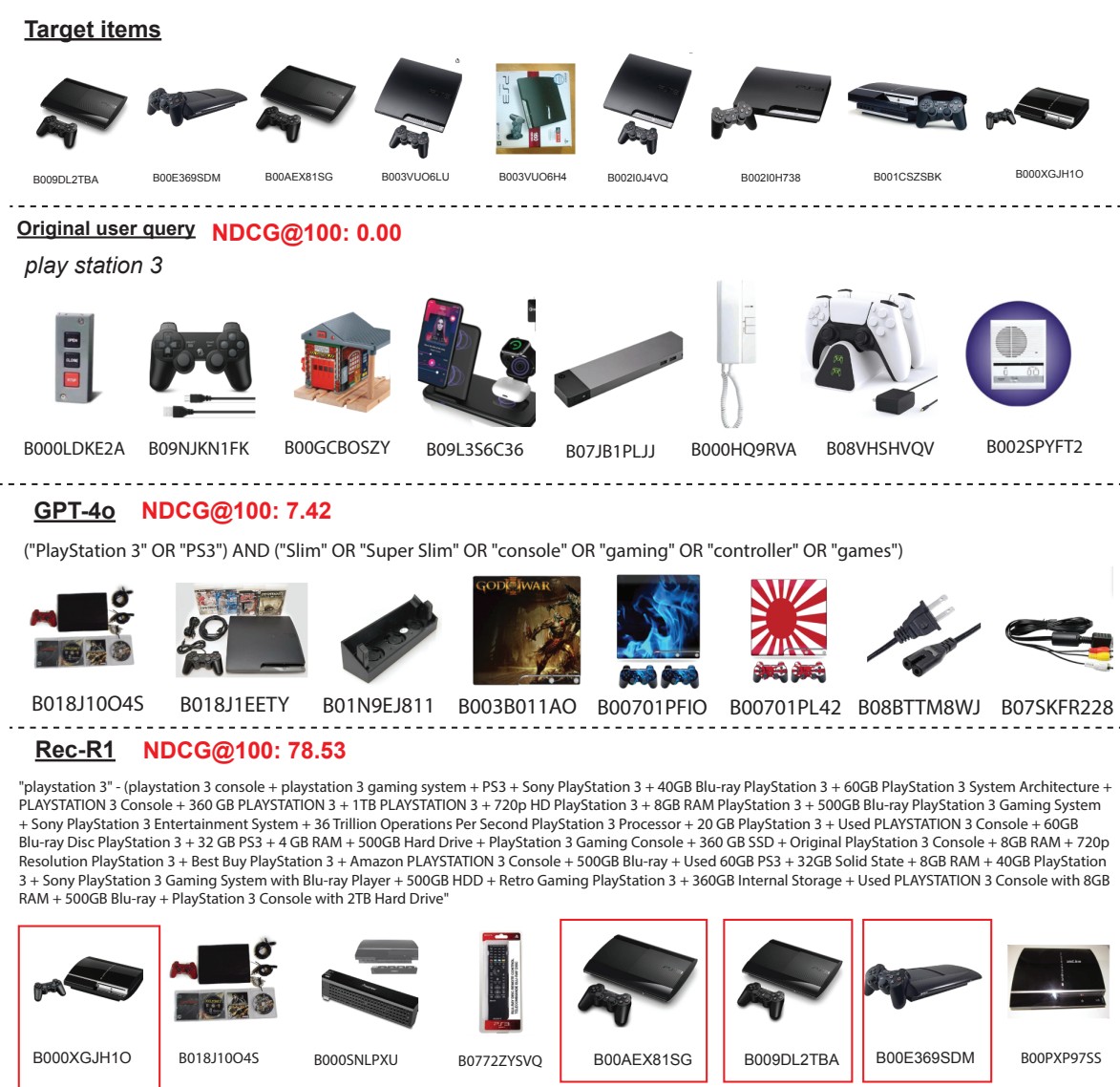

Figure 5: Qualitative comparison of retrieval results on the ESCI **Video Games** domain. We visualize the top-8 items retrieved by BM25 using different query formulations: the original user query (*play station 3*), a rewritten query by GPT-4o, and the output of our REC-R1. Ground-truth relevant items are shown at the top. REC-R1 significantly improves NDCG@100 by generating a highly detailed and semantically rich query, enabling precise matching with relevant items. Items correctly retrieved (i.e., appearing in the target set) are highlighted with red bounding boxes.

### G.1 Product Search: ESCI

Figure 5 illustrates a query rewriting example from the ESCI *Video Games* domain. The user's original query is simply "play station 3", which fails to retrieve any relevant items using BM25 (NDCG@100 = 0.00). When using GPT-4o for rewriting, the generated Boolean query covers many relevant keywords (e.g., "Slim", "controller", "games") but lacks grounding in product-specific terminology, leading to limited improvement (NDCG@100 = 7.42).

In contrast, REC-R1 generates a highly detailed and context-rich query that resembles human-level product search behavior. It incorporates specific configurations (e.g., storage size, "Blu-ray", "used" vs. "new"), product features, and variant types, all while maintaining semantic coherence. This significantly improves retrieval accuracy, correctly retrieving most of the target items (NDCG@100 = 78.53). Items that hit the target list are marked with red boxes in Figure 5, clearly demonstrating REC-R1 's superiority in aligning generation with downstream relevance signals.

### G.2 Sequential Recommendation: Amazon Beauty

Figure 6 shows a case from the Amazon Beauty dataset. The user previously purchased a set of products. The goal is to predict the next item the user is likely to purchase. GPT-4o generates a broad and somewhat vague Boolean query with high-level categories such as "makeup brushes" or "serum", failing to capture the continuity in user intent. As a result, none of the retrieved items match the target (NDCG@10 = 0.00).

REC-R1, on the other hand, generates a natural-sounding pseudo-review that maintains semantic consistency with the user's history while narrowing in on likely next items—such as winged eyeliner and related cosmetic tools. This leads to successful retrieval of the ground-truth target item (NDCG@10 = 43.06), as highlighted in red.

**History Items**

B01GS8J18E — B085ZXRQ1D (1 Pack)

**Target**

2 Pack — B085ZTR42P

**GPT-4o**
**NDCG@10: 0.00**

("hair mask" OR "styling gel" OR "hair oil") OR ("mascara" OR "eyeshadow" OR "eyebrow kit") OR ("facial cleanser" OR "moisturizer" OR "serum") OR ("makeup brushes" OR "hair styling tools")

B085MRMX9K  B087NMGGRV  B07N8VSYHN  B07H9823TH  B01GZV7MG2

**Rec-R1**
**NDCG@10: 43.06**

Winged eyeliner stamp black liner Perfect Cat Eye Makeup for Moroccan Argan Oil Hair Treatment Gift Set from Moroccan Argan Oil Shampoo and conditioner damaged hair repair Winged liner tips from Leila using natural products Wingliner from Perfect 2 in 1 dual end eyeliner brushes

B08DKG57TP  B08DKL47BR  B085ZXRQ1D (1 Pack)  B085ZTR42P (2 Pack)  B07JQGD781

Figure 6: Case study on the **Amazon Beauty** dataset for the sequential recommendation task. Given a user's historical purchase sequence (top-left), we compare the top-5 items retrieved by BM25 when queries are generated by GPT-4o (top) and Rec-R1 (bottom). Ground-truth target items are shown at the top-right. Rec-R1 produces a more contextually relevant and descriptive query, enabling accurate retrieval of both target items, significantly outperforming GPT-4o. Items correctly retrieved are highlighted with red bounding boxes.

