# OpenReview forum: "Rec-R1: Bridging Generative Large Language Models and User-Centric Recommendation Systems via Reinforcement Learning"
_TMLR — Accepted by TMLR_

### Review · Reviewer_DH2W · 2025-08-01

**Summary Of Contributions:**

The authors hook-up a tunable LLM into some tasks the require text (query-rewriting, mostly) and the train it with RL using rewards computed from the end-task (e.g. NDCG). Seems to work well!

**Audience:**

Yes

**Audience Explanation:**

LLMs + RL + retrieval -> what's not to love?

**Claims And Evidence:**

No

**Claims Explanation:**

The paper looks good in terms of results on the two recommendation tasks—no concerns there. But the overall framing needs adjustment. Your method essentially tackles a "query rewriting/expansion" problem, making it applicable to retrieval systems in general. Given this, you should be comparing your approach against established query rewriting baselines like Doc2Query, DocT5Query (and their modern SOTA brethren), and evaluating it on standard IR benchmarks like TREC-DL, MS MARCO, and BEIR.

Right now, the two recommendation datasets you're using function essentially as retrieval tasks, which means you're doing query reformulation/generation/rewriting + retrieval. But you're claiming this as a general user-centric recommender system framework, which feels overstated. If you truly want to position this as a broad RecSys method, you should show that it also works outside of retrieval-only scenarios. For instance, how would this method integrate with classic collaborative filtering tasks (Netflix Prize, Movielens), where inputs are user/item IDs instead of textual queries? Or do you think this doesn't count as "user centric"? If so, what DOES and how is it different from just retrieval?

What about the literature? Has anyone else used RL to teach LMs to rewrite queries? This is a pretty interesting and timely topic...

**Requested Changes:**

Strongly recommended: reframe the paper as being about RL-learned query rewrting.
IF you choose not to reframe the paper, then (required for acceptance): provide a complete discussion on which kinds of recsys datasets/problems your approach applies to and which ones it does not.

Required for acceptance: compare to SOTA query rewriting methods on DL19, DL20 and BEIR.

---

> ### Author Response · Authors · 2025-08-21
> **Response by the Authors**
>
> We thank the reviewer for the time investment and valuable feedback. We would like to clarify that our work is not intended to be positioned merely as another “query rewriting/expansion” approach. Instead, our primary focus is on the broader problem of **LLM4Rec**, which has recently emerged as a highly active and timely research topic RecSys communities, as shown by the recent comprehensive survey by Lin et al.
>
> As summarized in this survey and also in Figure 4 of our paper, LLM4Rec can take several major forms—including **query rewriting, feature augmentation, closed-set item generation, and open-set item generation**. Our work contributes a general reinforcement learning framework (Rec-R1) that is compatible with these paradigms, and our experiments validate its effectiveness in three of them.
>
> Traditional approaches typically rely on supervised fine-tuning (SFT) of LLMs, but SFT data often falls short of optimal quality, because the best possible supervision can only be revealed through interaction with the recommendation system. For example, in a product search scenario, without interacting with the RecSys, we cannot know what constitutes a truly better reformulation of the query. Once LLMs are allowed to interact, downstream metrics such as NDCG or Recall naturally serve as feedback signals that indicate the quality of generated text. **This creates a natural bridge for using Reinforcement learning with verifiable reward (RLVR) to align LLMs with RecSys objectives.**
>
> > Lin, Jianghao, et al. "How can recommender systems benefit from large language models: A survey." ACM Transactions on Information Systems 43.2 (2025): 1-47.
>
> ---
>
> Below, we address the comments by the reviewer.
>
> 1. Scope of Applicability
>
> We thank the reviewer for pointing out the importance of clarifying applicability. **We will provide a more complete discussion on which types of recommendation datasets/problems our approach applies to**. In particular, Rec-R1 focuses on tasks under the LLM4Rec setting, i.e., those where text can serve as a natural bridge between LLMs and RecSys signals.
>
> Regarding the reviewer’s question on collaborative filtering tasks: to the best of our knowledge, collaborative filtering is a methodology designed to predict new user–item interactions based on existing interaction histories. In fact, our sequential product recommendation task falls under this category. However, instead of relying on classic collaborative filtering techniques, we focus on semantic level, where item sequences are represented and processed in textual form. This corresponds to the fourth paradigm (**open-set item generation**) in the taxonomy of LLM4Rec and thus we follow such modeling. Within this setting, our proposed framework Rec-R1 can be directly applied to optimize LLM generations with RecSys feedback, thereby demonstrating its applicability beyond pure retrieval scenarios.
>
> 2. On query rewriting baselines
>
> We thank the reviewer for raising this important point. In response, we compared our approach against DocT5Query, a representative baseline from the Doc2Query family. The results on ESCI + BM25 and Amazon-C4 + BM25 are shown below:
>
> **ESCI + BM25**
>
> | Model     | Video Games | Baby Products | Office Products | Sports and Outdoors |
> |-----------|-------------|---------------|-----------------|---------------------|
> | DocT5Query | 15.02       | 14.37         | 17.19           | 16.83               |
> | Rec-R1     | 33.89       | 29.27         | 34.61           | 31.92               |
>
> **Amazon-C4 + BM25**
>
> | Model     | Video Games | Baby Products | Office Products | Sports and Outdoors |
> |-----------|-------------|---------------|-----------------|---------------------|
> | DocT5Query | 3.90        | 5.81          | 4.93            | 6.43                |
> | Rec-R1     | 26.51       | 27.04         | 23.10           | 27.40               |
>
> As can be seen, Rec-R1 significantly outperforms DocT5Query across all categories and datasets. This is expected, since Rec-R1 leverages interactive optimization via RLVR, enabling the LLM to gradually improve itself toward generating high-quality queries while suppressing low-quality ones, as reflected in the much higher NDCG scores.

---

> > ### Author Response · Authors · 2025-08-21
> > **Response by the authors**
> >
> > 3. On evaluation with IR benchmarks
> >
> > We also conducted evaluations on standard IR benchmarks, including MS MARCO and two representative datasets from BEIR (NFCorpus and FEVER). The results are presented below:
> > | Model              | MS MARCO | NFCorpus | FEVER |
> > |--------------------|----------|----------|-------|
> > | BM25               | 44.80    | 14.67    | 44.25 |
> > | Qwen2.5-3B-Instruct| 23.00    | 29.23    | 46.68 |
> > | GPT-4o             | 39.43    | 30.67    | 53.94 |
> > | Rec-R1             | 53.13    | 34.00    | 66.38 |
> >
> > As can be seen, Rec-R1 consistently outperforms the baselines across all datasets, further validating its effectiveness as a retrieval optimization framework. Even so, we would like to emphasize that **retrieval is not the main focus of our paper**. **Our contribution lies in the broader scope of LLM4Rec**, where the key challenge is how to bridge LLMs with RecSys signals in a general and principled way. The retrieval setting serves as one instantiation of this idea, but our framework is designed to extend beyond retrieval-only scenarios.
> >
> >
> >
> >
> > 4. On related literature
> >
> > We thank the reviewer for raising this question. As noted in Appendix B, our submission already discusses related work on Generative LLMs for Recommendation Tasks and Reinforcement Learning for Recommendation Systems. In the final version, we will further expand the related work section to explicitly include LLM for query rewriting.
> >
> > In particular, as shown in the following relevant directions:
> > * LLM for query rewriting (zero-shot or SFT): Gao et al., Khattab et al.,  Ye et al., Mackie et al. and Liu et al. proposed leveraging LLM capabilities for query rewriting in retrieval tasks.
> > * RL-based query reformulation: Prior works such as Wu et al., Nogueira et al., Chen et al.,  applied RL to optimize query rewriting for search.
> >
> > While these works explore using RL for query reformulation, there are several key drawbacks compared with Rec-R1:
> > * Closed-loop recommendation feedback: Prior work often relies on synthetic or proxy rewards, whereas our approach directly uses evaluation metrics (e.g., NDCG, Recall) from a downstream recommendation model as reward signals—thus forming a true closed-loop interaction between the LLM and RecSys.
> > * Model scale and capability: These works use relatively weak models before the LLM era without strong generalization or reasoning ability. In contrast, our goal is to adapt powerful open-domain LLMs to recommendation scenarios—precisely because traditional RecSys lacks such capabilities, as mentioned in the Introduction.
> > * Task scope: Compared with Rec-R1, these papers do not focus on recommendation systems, but more narrowly on search query refinement. Our framework is designed to support diverse tasks within RecSys.
> >
> > We once again thank the reviewer for their valuable suggestions and constructive feedback. We will carefully incorporate these insights to further improve the clarity, completeness, and overall quality of our final version.
> >
> > **References**
> >
> > * Gao, Luyu, et al. "Precise zero-shot dense retrieval without relevance labels." Proceedings of the 61st Annual Meeting of the Association for Computational Linguistics (Volume 1: Long Papers). 2023.
> > * Khattab, Omar, and Matei Zaharia. "Colbert: Efficient and effective passage search via contextualized late interaction over bert." Proceedings of the 43rd International ACM SIGIR conference on research and development in Information Retrieval. 2020.
> > * Ye, Fanghua, et al. "Enhancing Conversational Search: Large Language Model-Aided Informative Query Rewriting." The 2023 Conference on Empirical Methods in Natural Language Processing.
> > * Mackie, Iain, Shubham Chatterjee, and Jeffrey Dalton. "Generative relevance feedback with large language models." Proceedings of the 46th international ACM SIGIR conference on research and development in information retrieval. 2023.
> > * Liu, Linqing, et al. "Query expansion using contextual clue sampling with language models." arXiv preprint arXiv:2210.07093 (2022).
> > * Wu, Zeqiu, et al. "CONQRR: Conversational Query Rewriting for Retrieval with Reinforcement Learning." Proceedings of the 2022 Conference on Empirical Methods in Natural Language Processing. 2022.
> > * Nogueira, Rodrigo, and Kyunghyun Cho. "Task-oriented query reformulation with reinforcement learning." 2017 Conference on Empirical Methods in Natural Language Processing, EMNLP 2017. Association for Computational Linguistics (ACL), 2017.
> > * Chen, Zhiyu, et al. "Reinforced Question Rewriting for Conversational Question Answering." Proceedings of the 2022 Conference on Empirical Methods in Natural Language Processing: Industry Track. 2022.

---

> > > ### Comment · Reviewer_DH2W · 2025-08-25
> > >
> > > Thanks for the detailed response and for adding DocT5Query and the MS MARCO / BEIR experiments.
> > >
> > > I acknowledge your framing via Lin et al.’s taxonomy (e.g., “Query Rewriting” vs. “Open-Set Item Generation”). I also agree that sequential recommendation from histories is not literally *query rewriting*. That said, my fundamental concern remains: in all instantiated settings, the LLM produces text that is then consumed by a retriever. Whether the text is a rewritten query or a generated description from history, the resolution mechanism is identical and retrieval-centric (“LLM-text -> retriever”).
> > >
> > > This matters for positioning. To claim Rec-R1 as a general LLM4Rec framework rather than a retrieval-mediated query-generation framework, I’d expect at least one application that does not reduce to “text -> retriever.”
> > >
> > > Concretely, please help me understand:
> > >
> > > 1. Definition boundary. What is the precise technical distinction (in modeling and training) between LLM4Rec, as used here, and LLM query-generation/rewriting + retrieval when the LLM output is always text passed to a black-box retriever?
> > >
> > > 2. Non-retrieval tasks. Do you have a task that is LLM4Rec but not retrieval-mediated, e.g.,
> > >    – Closed-set item ranking where the LLM directly scores a fixed candidate set (no text retrieval in the loop), or
> > >    – Direct item-ID selection via constrained decoding over item vocabularies, or
> > >    – LLM re-ranking of provided candidates *without* going back through a retriever, with Rec-R1 rewards defined on Recall/NDCG of that candidate set?
> > >    If such tasks fit Rec-R1, please include one to substantiate generality beyond retrieval.
> > >
> > > 3. If not instantiated here, please either (a) add one non-retrieval experiment (even small-scale) or (b) scope the claims to retrieval-mediated settings.
> > >
> > > On evaluation: thanks for the additional baselines/results. To meet the bar I set earlier, I still need:
> > > - DL’19 and DL’20 results; and
> > > - Comparisons against strong, modern query-expansion/rewriting baselines beyond DocT5Query (recent LLM-assisted QE/GenQ/GRF-style methods, etc.).
> > >
> > > These are necessary to substantiate superiority in the retrieval-oriented instantiations you currently present.
> > >
> > > If you can (i) clearly delineate the boundary between “general LLM4Rec” and “text-to-retriever,” and (ii) include at least one non-retrieval Rec-R1 instantiation or narrow the claims accordingly, plus (iii) the missing benchmarks/baselines above, I’d view the positioning as resolved.
> > >
> > > Context vs. RL-for-search lines (required comparison):
> > > Please contextualize Rec-R1 against closely related RL-for-search approaches such as Search-R1 and ConvSearch-R1, which train LLMs via RL to generate search queries and reason with multi-turn retrieval using simple outcome rewards. These works report sizable gains on search/QA and CQR tasks, respectively, and release code/checkpoints. How does Rec-R1 differ technically (e.g., reward design tied to RecSys metrics like NDCG/Recall, training templates, interleaved control, stability) and applicability (datasets beyond QA/search)? If applicable, include them as baselines or explain why a direct comparison is not feasible.

---

> > > > ### Author Response · Authors · 2025-08-27
> > > > **Response by the Authors**
> > > >
> > > > We thank the reviewer for the time investment and valuable feedback. Below, we address the follow-up questions by the reviewer.
> > > >
> > > > 1. We thank the reviewer for the constructive suggestions regarding the framing of our paper. To address the concern about whether Rec-R1 applies beyond retrieval settings, we have added a **Closed-Set Item Ranking** experiment (i.e., product item reranking).
> > > >
> > > > The input consists of a user query and $m$ candidate items (each represented by text descriptions, here m=16). The LLM is required to produce a ranked list of the candidate items. During training, we compute the NDCG score of the ranked list against the ground-truth and use this as the reward for Rec-R1 optimization.
> > > >
> > > > The results are summarized below:
> > > >
> > > > | Model                | Video Games | Baby Products | Office Products | Sports and Outdoors |
> > > > |----------------------|-------------|---------------|-----------------|---------------------|
> > > > | Qwen3-Reranker-4B    | 44.29       | 46.41         | 48.06           | 47.42               |
> > > > | Qwen3-Reranker-8B    | 44.28       | 44.95         | 47.73           | 44.51               |
> > > > | Qwen2.5-3B-Instruct  | 5.38        | 9.21          | 4.26            | 4.37                |
> > > > | GPT-4o               | **61.57**       | 56.18         | 53.62           | 53.62               |
> > > > | Rec-R1-3B            | 53.04   | **57.08**     | **60.91**       | **62.91**           |
> > > >
> > > > From the table, Rec-R1 consistently achieves the best overall performance. Notably, compared with the initialized Qwen2.5-3B model, Rec-R1 delivers very large improvements. Furthermore, against SoTA open-source embedding/reranker models (Qwen3-Reranker family), Rec-R1 demonstrates substantial superiority.
> > > >
> > > > These results confirm that Rec-R1’s high-level optimization framework is **not limited to retrieval tasks**. Instead, it applies naturally to closed-set item ranking, supporting our original claim that Rec-R1 provides a general way to **bridge LLMs and RecSys with RLVR**.
> > > >
> > > > Therefore, regarding the boundary between “general LLM4Rec” and “LLM query-generation/rewriting + retrieval,” our view is that query rewriting is **one specific subcategory** of LLM4Rec, as also summarized in **Figure 4** and consistent with the taxonomy in Lin et al. (2025) and the recommendation system community. We will revise the paper to make this positioning clearer.
> > > >
> > > > We thank the reviewer again for these valuable suggestions, which helped us strengthen both the empirical scope and the conceptual clarity of our work.
> > > >
> > > > 2.  Comparisons with Modern Strong Query-Rewriting Baselines
> > > >
> > > > We thank the reviewer for pointing this out. We further compared Rec-R1 against several recent and strong LLM-assisted query rewriting/expansion baselines (beyond DocT5Query) pulished at top venues, including RetPO, LLM4CS, and AdaQR, on ESCI and Amazon-C4.
> > > >
> > > > The results are shown below:
> > > >
> > > > ESCI
> > > >
> > > > | Model       | Video Games | Baby Products | Office Products | Sports and Outdoors |
> > > > |-------------|-------------|---------------|-----------------|---------------------|
> > > > | DocT5Query  | 15.02       | 14.37         | 17.19           | 16.83               |
> > > > | RetPO       | 23.43       | 17.88         | 19.00           | 20.80               |
> > > > | LLM4CS      | 23.29       | 24.02         | 28.06           | 28.00               |
> > > > | AdaQR       | 23.53       | 20.59         | 29.02           | 26.36               |
> > > > | **Rec-R1**  | **33.89**   | **29.27**     | **34.61**       | **31.92**           |
> > > >
> > > > Amazon-C4
> > > >
> > > > | Model       | Video Games | Baby Products | Office Products | Sports and Outdoors |
> > > > |-------------|-------------|---------------|-----------------|---------------------|
> > > > | DocT5Query  | 3.90        | 5.81          | 4.93            | 6.43                |
> > > > | RetPO       | 5.20        | 1.40          | 1.64            | 2.54                |
> > > > | LLM4CS      | 13.05       | 9.97          | 11.51           | 11.05               |
> > > > | AdaQR       | 12.98       | 9.15          | 10.47           | 10.41               |
> > > > | **Rec-R1**  | **26.51**   | **27.04**     | **23.10**       | **27.40**           |
> > > >
> > > > As shown, Rec-R1 consistently outperforms these strong LLM-assisted baselines by a substantial margin across all domains.
> > > >
> > > > * Mao, Kelong, et al. "Large Language Models Know Your Contextual Search Intent: A Prompting Framework for Conversational Search." The 2023 Conference on Empirical Methods in Natural Language Processing.
> > > > * Yoon, Chanwoong, et al. "Ask Optimal Questions: Aligning Large Language Models with Retriever’s Preference in Conversation." Findings of the Association for Computational Linguistics: NAACL 2025. 2025.
> > > > * Zhang, Tianhua, et al. "Adaptive Query Rewriting: Aligning Rewriters through Marginal Probability of Conversational Answers." Proceedings of the 2024 Conference on Empirical Methods in Natural Language Processing. 2024.

---

> > > > > ### Author Response · Authors · 2025-08-27
> > > > > **Response by the Authors**
> > > > >
> > > > > 3. On evaluation with IR benchmarks
> > > > >
> > > > > To further strengthen our evaluation, we additionally report results on the DL’19 and DL’20.
> > > > >
> > > > > | Model               | DL’19  | DL’20  |
> > > > > |---------------------|--------|--------|
> > > > > | BM25                | 42.31 | 47.80 |
> > > > > | Qwen2.5-3B-Instruct | 43.56 | 52.24 |
> > > > > | GPT-4o              | 59.87 | 54.72 |
> > > > > | **Rec-R1**          | **63.25** | **59.71** |
> > > > >
> > > > > As shown, Rec-R1 achieves the best overall performance on both DL’19 and DL’20. This further validates the effectiveness of Rec-R1 under standard IR benchmarks.
> > > > >
> > > > > 4. Compared with Search-R1 and ConvSearch-R1
> > > > >
> > > > > We thank the reviewer for pointing out these two related work. We will add them to the Related Work section. The reason they were not included earlier is that Search-R1 and Rec-R1 were concurrent works (both in March), while ConvSearch-R1 appeared later (May). Below we provide a detailed comparison.
> > > > >
> > > > > **Search-R1.**
> > > > > As described in Section 3.4 of their paper, Search-R1 relies solely on a single *exact matching reward*. In contrast, Rec-R1 directly optimizes downstream recommendation metrics (e.g., Recall, NDCG) as the reward function. This distinction is important: Search-R1 cannot explicitly constrain the quality of the intermediate retrieval process due to the property of RLVR optimization paradigm, while Rec-R1 is directly aligned with the final RecSys evaluation objectives.
> > > > >
> > > > > We also compared Search-R1 and Rec-R1 on ESCI and Amazon-C4 datasets. Results are shown below:
> > > > >
> > > > > ESCI
> > > > >
> > > > > | Model       | Video Games | Baby Products | Office Products | Sports and Outdoors |
> > > > > |-------------|-------------|---------------|-----------------|---------------------|
> > > > > | Search-R1   | 18.03       | 17.38         | 22.67           | 19.62               |
> > > > > | **Rec-R1**  | **33.89**   | **29.27**     | **34.61**       | **31.92**           |
> > > > >
> > > > > Amazon-C4
> > > > >
> > > > > | Model       | Video Games | Baby Products | Office Products | Sports and Outdoors |
> > > > > |-------------|-------------|---------------|-----------------|---------------------|
> > > > > | Search-R1   | 12.06       | 9.12          | 10.21           | 9.96                |
> > > > > | **Rec-R1**  | **26.51**   | **27.04**     | **23.10**       | **27.40**           |
> > > > >
> > > > > As shown, Rec-R1’s direct reward optimization leads to significantly better performance across domains.
> > > > >
> > > > > **ConvSearch-R1.**
> > > > > ConvSearch-R1 differs in its reward design: instead of directly optimizing evaluation metrics, it introduces a piecewise-defined reward function (Eq. (1) in their paper). However, upon careful analysis, we believe some of their claims deserve further discussion. Specifically, their motivation using Eq. (1) reward function is that using NDCG@3 as a reward would cause reward sparsity. Our comment: this sparsity is expected if one insists on optimizing only at k=3. In fact, it is easy to derive that optimizing at larger k implicitly improves smaller-k NDCG as well. Moreover, ConvSearch-R1’s reward design does not directly align with evaluation metrics. From the results below, we found that Rec-R1 consistently outperform ConvSearch-R1.
> > > > >
> > > > > ESCI
> > > > >
> > > > > | Model       | Video Games | Baby Products | Office Products | Sports and Outdoors |
> > > > > |-------------|-------------|---------------|-----------------|---------------------|
> > > > > | ConvSearch-R1 | 23.67     | 22.35         | 27.89           | 25.64               |
> > > > > | **Rec-R1**  | **33.89**   | **29.27**     | **34.61**       | **31.92**           |
> > > > >
> > > > > Amazon-C4
> > > > >
> > > > > | Model       | Video Games | Baby Products | Office Products | Sports and Outdoors |
> > > > > |-------------|-------------|---------------|-----------------|---------------------|
> > > > > | ConvSearch-R1 | 13.57     | 13.08         | 13.62           | 13.34               |
> > > > > | **Rec-R1**  | **26.51**   | **27.04**     | **23.10**       | **27.40**           |
> > > > >
> > > > > In summary, Rec-R1 distinguishes itself by (i) reward design directly tied to RecSys evaluation metrics, (ii) consistently stronger empirical performance on LLM4Rec tasks.
> > > > >
> > > > > ---
> > > > >
> > > > > We sincerely thank the reviewer again for the constructive feedback. We will promptly update the paper to (i) include these baselines and discussions, (ii) clarify the boundary between general LLM4Rec and query-rewriting, and (iii) incorporate the new experiments (closed-set item ranking, DL’19/20, and strong LLM-assisted QE baselines).
> > > > >
> > > > > We hope these updates resolve the reviewer’s concerns. Of course, if there remain further points of clarification, we would be glad to continue the discussion.

---

> > > > > > ### Comment · Reviewer_DH2W · 2025-08-27
> > > > > >
> > > > > > Thank you, that addresses my concerns.
> > > > > >
> > > > > > Assuming all of these benchmarks appear in the paper, I am satisfied.

---

> > > > > > > ### Author Response · Authors · 2025-08-28
> > > > > > > **Response by the Authors**
> > > > > > >
> > > > > > > We sincerely thank the reviewer for the positive feedback. We have carefully revised the manuscript and incorporated all the discussed changes. In particular, all the additional benchmarks have now been included: Product Re-ranking (Table 4), Standard IR datasets (Table 17), stronger modern query-rewriting baselines (Tables 11 and 12), as well as direct comparisons with Search-R1 and ConvSearch-R1 (Tables 18 and 19).
> > > > > > >
> > > > > > > We greatly appreciate the reviewer’s time and thoughtful comments, which have significantly improved the clarity and completeness of our paper.

---

### Review · Reviewer_MD4W · 2025-08-13

**Summary Of Contributions:**

The paper studies ways LLMs have been used in recommendation systems. They focus specifically on the setting where there is an existing recommendation system that turns natural language input into a user recommendation.

Prior methods have tried using LLMs to transform initial natural language input into queries with more information, either using prompted versions of the LLM or applying SFT to the LLMs. This paper instead applies RL, using the reward from the recommendation system as the target.

There is nothing particularly wrong with the paper's experiments or problem setups, showing that a GRPO based RL training method can optimize the recommendation system better than an SFT setup can, and that RL can outperform the data in SFT generated by a fixed expert (in this case, GPT-4o). I mostly find myself skeptical that this is of value, given that

1) It has been shown outside of recommendation systems that RL can be more effective than SFT at satisfying user preferences, across a wide swath of LLM work, including the very first InstructGPT papers.

2) It has been shown that RL can be an effective means of optimizing reward within recommendation systems, for different deep learning architectures.

So it's not clear to me where the value of the paper is. I also think that it's a bit silly to include a proof that the MLE objective minimizes the KL divergence between the learned policy and a fixed target policy - this is just a well-known result.

Within Appendix B, the authors make the case for related work, arguing that prior applications of LLMs to recommendation systems either used an offline reward model rather than the true environment, or used DPO, or used more narrow forms of a recommendation system environment. I'm not sure these are meaningful enough differences.

As for the estimated cost breakdown, where the authors argue the training of the model runs at 1/30th of the cost of the SFT pipeline, I agree with the numbers as stated, but it is worth mentioning that the Rec-R1 proposal would also involve regret from the currently learning model (the $ lost from exploring different recommendations than the baseline model, from the live recommendation system setup).

**Audience:**

No

**Audience Explanation:**

As mentioned earlier, it is unclear to me that a paper showing RL outperforms SFT for LLMs is of interest, when this has been shown for LLMs in wider settings, and when it has also been shown in recommendation systems without LLMs. That it works in the intersection seems clear to me.

**Claims And Evidence:**

Yes

**Claims Explanation:**

Experimental setups seem reasonable, with ablations done across model architectures, different recommendation system settings, etc.

**Requested Changes:**

Most of my reservations about the work are at the structural level, rather than content level, so I'm unsure there's any adjustment that would affect my recommendation. I would need to see a more convincing argument that this paper says something of interest to the wider ML community.

---

> ### Author Response · Authors · 2025-08-21
> **Response by the Authors**
>
> We respectfully disagree with the reviewer’s concern regarding the value of our work. The contribution of our paper is not a trivial restatement that “RL can outperform SFT,” but rather the demonstration of **a general closed-loop reinforcement learning framework (Rec-R1) for bridging LLMs and RecSys, focusing on LLM4Rec**, which has recently emerged as a highly active and timely research topic RecSys communities, as shown by the recent comprehensive survey by Lin et al.
>
> We would like to highlight that both other reviewers emphasized our paper's scope and its importance:
> * **Reviewer DH2W** explicitly noted that “LLMs + RL + retrieval → what’s not to love?”, claiming that our work focuses on a “pretty interesting and timely topic.”
> * **Reviewer Q5qx** similarly recognized that “TMLR's audience would likely be interested in this paper's approach of using reinforcement learning to connect LLMs and recommendation systems via feedback optimization,” further observing that “Rec-R1 has attracted considerable attention since its release as a preprint.”
>
> These remarks underscore that the contribution is not about re-proving a well-known principle of RL > SFT, but about focusing on how to bridge LLMs and RecSys within LLM4Rec environments—where language models serve as semantic mediators to interact with black-box recommenders, and rewards are derived from downstream recommendation metrics (NDCG, Recall) rather than human preference labels or synthetic proxies.
>
> To be more specific, as summarized in this survey and also in Figure 4 of our submission, LLM4Rec can take several major forms—including query rewriting, feature augmentation, closed-set item generation, and open-set item generation. Our work contributes a general reinforcement learning framework (Rec-R1) that is compatible with these paradigms.
>
> Traditional approaches typically rely on supervised fine-tuning (SFT) of LLMs, but SFT data often falls short of optimal quality, because the best possible supervision can only be revealed through interaction with the recommendation system. For example, in a product search scenario, without interacting with the RecSys, we cannot know what constitutes a truly better reformulation of the query. Once LLMs are allowed to interact, downstream metrics such as NDCG or Recall naturally serve as feedback signals that indicate the quality of generated text. **This creates a natural bridge for using Reinforcement learning with verifiable reward (RLVR) to align LLMs with RecSys objectives.**
>
>
> > Lin, Jianghao, et al. "How can recommender systems benefit from large language models: A survey." ACM Transactions on Information Systems 43.2 (2025): 1-47.
>
>
> ---
>
> Below, we address the comments by the reviewer.
>
> 1. Related to the Proof
>
> We respectfully disagree with the reviewer’s characterization of Theorem 1. While the fact that the MLE objective minimizes KL divergence may indeed be considered well-known by you and by us, it is by no means universally familiar to all researchers and practitioners in the recommender systems community—who represent a large portion of our intended audience. Our inclusion of the proof serves an important purpose: **it ensures that the paper is self-contained and makes explicit why SFT inherently suffers from a “performance ceiling”.**
>
> By presenting this result formally, we provide clarity for readers who may not have the background and make transparent the theoretical motivation for introducing Rec-R1. In particular, this bridge helps connect SFT’s limitation (only imitating a fixed distribution) with the need for reinforcement learning to leverage interactive feedback from RecSys environments. Thus, while the proof may be familiar to some, we believe it is necessary and valuable in the context of this work.
>
> 2. On the Concern of Exploration Cost
>
> We respectfully clarify that the concern about potential “regret” from exploration is not that much in our setting. In Rec-R1, the model is optimized with the most recent historical interaction data, meaning that it does not need to “try out” poor recommendations on live users in order to improve. Consequently, there is not that much monetary or engagement loss from exploration.
>
> ---
>
> We thank the reviewer for their comments. While we understand the skepticism, we note that other reviewers explicitly highlighted the value of our work focusing on LLM4Rec and recognized Rec-R1’s contribution as a meaningful step in this area. We will make sure our final version emphasizes this broader positioning so that the value of the work is clear to all readers.

---

> ### Author Response · Authors · 2025-09-13
> **Invitation for Discussion**
>
> Dear Reviewer MD4W,
>
> We sincerely appreciate the time and effort you have dedicated to reviewing our manuscript. During the recent discussion period, we carefully considered the comments from all reviewers and updated our paper accordingly. The other two reviewers (**Reviewer DH2W** and **Reviewer Q5qx**) have also provided positive feedback on the revised version.
>
> We would like to kindly check whether you might have any additional questions or comments regarding our work.
>
> Thank you again for your thoughtful review and engagement.
>
> Best regards,
>
> Authors

---

### Review · Reviewer_Q5qx · 2025-08-13

**Summary Of Contributions:**

Rec-R1 is a closed-loop reinforcement learning framework that connects LLMs with recommender systems, directly optimizing the generation policy using recommendation metrics. Experiments on product search and sequential recommendation show that Rec-R1 outperforms prompting and SFT approaches while preserving general knowledge and reasoning abilities. The authors demonstrate the broad applicability of different recommender architectures and tasks.

I list key strengths and weaknesses below:

Strengths:
- The authors effectively use Figure 2 to establish a clear position within existing LLM-based recommendation system methods, effectively communicating their approach to readers.
- The authors attempt to theoretically support that Rec-R1 can overcome SFT's limitations through Theorem 1. By showing SFT's 'performance ceiling' via Theorem 1, they tried to demonstrate why reinforcement learning is necessary, and empirically show the limitations of SFT through Figure 1(a).
- The method performs well in cold-start scenarios where user profile information is absent.

Weaknesses:
- While Theorem 1 attempts to reveal the research motivation and SFT's limitations, the three assumptions required for this may not be realistic.
- Although the title mentions "user-centric" recommendation, the proposed method and experiments are closer to rule-based RL that uses static offline metrics as rewards.
- Rec-R1 only uses a single reward based on final recommendation quality metrics (e.g., NDCG, Recall). This may lead to insufficient feedback regarding diversity or format consistency.
- Experiments are limited to small-scale datasets.

**Additional Comments:**

- As mentioned in the introduction, since the proposed framework assumes RecSys as a "fixed black box," wouldn't it be difficult for the LLM to access RecSys's internal logic (e.g., lack of understanding user preferences, difficulty processing implicit feedback)?
- While the early parts of the paper use terms like "model-agnostic, task-flexible," I'm uncertain whether these terms apply throughout the entire paper. Given the prerequisite that rule-based reward computation must be possible, readers holding onto these terms might become confused as they read through the paper.

**Audience:**

Yes

**Audience Explanation:**

TMLR's audience would likely be interested in this paper's approach of using reinforcement learning to connect LLMs and recommendation systems via feedback optimization. In addition, Rec-R1 has attracted considerable attention since its release as a preprint.

**Claims And Evidence:**

No

**Claims Explanation:**

- The title and introduction prominently feature "user-centric" and "closed-loop," but the rewards consist of ranking metrics (e.g., NDCG and Recall) from static benchmarks rather than signals approximating actual user utility or long-term satisfaction. This may come across to readers as offline quality optimization rather than learning with user-centric signals.

- The authors use Figure 1 (a) to empirically support SFT's fundamental limitations. From this perspective, their claim is somewhat convincing.

- The proof in Appendix C.1 focuses only on SFT without directly comparing REC-R1's theoretical superiority. This may claim that RL "overcomes" SFT's limitations, but it appears to rely solely on the empirical results in Figure 1 for the readers.

**Requested Changes:**

- I believe it would be beneficial if the authors could additionally address the limitations of the three assumptions in Theorem 1. For example, 3B-sized or smaller LLMs may struggle to mimic the complex distribution of larger models like GPT-4o perfectly. If the learning model has insufficient capacity, the theorem's conclusion may not hold. Additionally, regarding the second assumption, a global optimum is rarely achieved in deep learning. For the third assumption, while the paper mentions "sufficient data," actual recommendation system data is noisy and sparse. Therefore, it seems unlikely to ideally reach the theorem's conclusion. A section or discussion addressing these limitations would be helpful.

- When introducing Equation (5), adding explicit differences from Equation (4) could clarify REC-R1's theoretical superiority. For example, adding a statement like "Unlike Equation (4), Equation (5) optimizes the true objective rather than a proxy..." would help readers understand better.

- The results in Table 3's transductive setting, where Rec-R1 lags behind some baselines like UniSRec, need further analysis and explanation.

- From the perspective of Rec-R1's reward structure, I'm curious whether it is less stable compared to the multi-reward structures proposed in Stream-Rec [1] or Search-R1 [2].

- Can the authors report learning and stability results under multi-objective rewards (accuracy, diversity, fairness, efficiency) rather than relying solely on recall and NDCG rewards?

> [1] Zhang, Junjie, et al. "Slow Thinking for Sequential Recommendation." arXiv preprint arXiv:2504.09627 (2025).
>
> [2] Jin, Bowen, et al. "Search-r1: Training llms to reason and leverage search engines with reinforcement learning." arXiv preprint arXiv:2503.09516 (2025).

---

> ### Author Response · Authors · 2025-08-21
> **Response by the Authors**
>
> We thank the reviewer for the thoughtful and constructive feedback. We are encouraged that the reviewer highlighted several strengths of our work, including: (i) the clear positioning of Rec-R1 within the landscape of LLM-based recommendation methods (Figure 2), (ii) the theoretical motivation provided by Theorem 1 in explaining SFT’s limitations and the necessity of RL, (iii) the strong empirical performance of Rec-R1 in cold-start scenarios, and (iv) the broad applicability of our framework across different recommendation architectures and tasks. We are glad that these contributions were recognized, as they capture the value of our work.
>
> ---
>
> We now address the reviewer’s concerns point by point.
>
> 1. On the Assumptions in Theorem 1
>
> We emphasize that the role of Theorem 1 is to formalize the **performance ceiling** of supervised fine-tuning (SFT). Specifically,
> - Under these assumptions, SFT is guaranteed to match the teacher policy but never surpass it.
> - When these assumptions are violated (e.g., insufficient model capacity, imperfect optimization, or noisy/sparse data), the learned policy will fall short of the teacher distribution, not exceed it.
>
> This means the theorem actually establishes a best-case scenario for SFT. In realistic recommendation environments where these assumptions rarely hold, SFT is even more limited. Thus, far from weakening our motivation, the reviewer’s concern actually reinforces our argument: if SFT is capped even under idealized conditions, its limitations are only more severe in practice (LLM4Rec scenarios), motivating the need for reinforcement learning.
>
> 2. On Assumption (i): Model Expressivity
>
> The reviewer raises the concern that a 3B-sized model may not have sufficient capacity to mimic the complex distribution of a larger teacher model such as GPT-4o. However, in our experiments, the 3B model already achieves very low training and validation loss when learning from the teacher’s outputs, indicating that it can fit the distribution effectively. While it is true that 3B parameters are smaller than GPT-4o, it is important to contextualize: 3B is still in the billion-scale LLM regime, orders of magnitude larger than NLP models before the LLM era with only millions of parameters.
>
> 3. On the Use of Static Metrics as Rewards
>
> We acknowledge the reviewer’s observation that our current experiments rely primarily on user history data and static metrics rather than direct online interactions with real users. We will revise the paper to make this distinction clearer. In our paper, we follow this setup because in some industry applications, it is common to do that.  By leveraging the most recent user history data, it can closely approximate real-world recommendation dynamics and align the model with the actual distribution of user behaviors and preferences. We will modify the paper to explicitly include these discussions to make our point clearer.
>
> 4. On the Transductive Setting Results of Sequential Recommendation
>
> We thank the reviewer for pointing out the performance gap in the transductive setting. We would like to note that we have already provided an extended discussion in Appendix E.2.4 (“Additional Analysis: Why Rec-R1 Performs Better in the Inductive Setting?”).
>
> As discussed there, the relatively lower performance of Rec-R1 in the transductive setting can be attributed to the fact that many test data overlap with training data. Traditional sequential recommendation models are typically trained from scratch or from weak embedding initialization to minimize prediction loss, and they can exploit this overlap by fitting co-occurrence patterns directly from large-scale data. In other words, once such models discover “shortcuts” that approximate frequent data associations, they can achieve strong performance simply by memorizing and reproducing these correlations.
>
> In contrast, Rec-R1 is designed to leverage semantic reasoning through language modeling, following the LLM4Rec paradigm (Figure 4). This requires the model to infer item transitions based on underlying intent or semantics rather than memorized co-occurrence. When the task lacks a strong logical mapping between history and future items, such reasoning becomes less effective and may even introduce noise.
>
> In the inductive setting, however, these memorization shortcuts are removed, forcing models to rely on transferable semantic patterns. This aligns more naturally with Rec-R1’s learning mechanism: the LLM is incentivized to generate generalized and meaningful reasoning process and optimize rewards accordingly. As a result, the inductive setting provides a clearer signal for reward-driven optimization, which explains why Rec-R1 outperforms baselines there.

---

> ### Author Response · Authors · 2025-08-21
> **Response by the Authors**
>
> 5. On Reward Structure and Multi-Objective Rewards
>
> We appreciate the reviewer’s insightful question on the stability of our reward design compared with Stream-Rec [1] and Search-R1 [2]. We would like to clarify that, from the perspective of Rec-R1’s framework, **our reward structure is fundamentally no different from those in existing work** (e.g., Stream-Rec). In Rec-R1, the reward is defined as a function f(a|s), and this formulation is general: f(a|s) can be extended to incorporate multiple reward signals by linearly combining them. In such a setup, the optimization naturally trains the model toward trajectories with higher overall aggregated rewards, which is the same principle used in Stream-Rec.
>
> In fact, our current implementation already follows a multi-reward design. Specifically, we combine two types of rewards:
> - Downstream metric reward (e.g., recall or NDCG), which measures recommendation quality.
> - Format reward, which ensures well-structured and valid outputs (this is also adopted by Stream-Rec).
>
> Our experiments show that this multi-reward setup yields stable training dynamics and strong performance, suggesting that Rec-R1 can effectively handle multiple reward signals.
>
> To further examine whether Rec-R1 remains stable under multi-objective rewards, we conducted additional experiments on the ESCI product search dataset. Following the multi-objective reward setting in Stream-Rec, we combined three types of rewards:  (1) format reward; (2) ndcg; (3) Category Consistency Reward, used for measuring the proportion of retrieved items that match the category of the target items given a user query.
>
> The results (shown below) demonstrate that **Rec-R1 trains stably under this multi-reward setting and continues to perform strongly** across different product domains:
>
> | Model  | Video Games | Baby Products | Office Products | Sports and Outdoors |
> |---|---|---|---|---|
> |Qwen-2.5-3B-Instruct|19.63|16.03|19.96|21.36|
> |GPT-4o|26.06|23.05|27.98|27.38|
> | Rec-R1 | 33.06 | 29.39 | 34.22| 32.27 |
>
> These findings indicate that Rec-R1 is not limited to single-metric optimization but can be readily extended to more complex multi-objective reward structures. We will incorporate these results and discussions into the revised version of the paper.
>
> Regarding the comparison to Search-R1 [2], we would like to note that Search-R1 does not adopt a multi-reward design. As described in Section 3.4 of their paper, they rely solely on a single exact matching reward, computed between the LLM’s answer and the ground-truth answer, without including any other rewards.
>
> 6. On Theoretical Comparison between SFT and RL
>
> We thank the reviewer for pointing out that the previous proof in Appendix C.1 only formalized the performance ceiling of SFT without explicitly comparing Rec-R1’s RL optimization with SFT. We have strengthened our theoretical analysis in the revision.
>
> - **Bound on SFT performance.** Lemma 1 Below establishes that the expected reward difference between the SFT policy $\pi_{\mathrm{SFT}}$ and the teacher policy $\pi_g$ is upper bounded by $\mathcal{O}(\sqrt{\kappa})$, where $\kappa$ is the expected KL divergence between $\pi_{\mathrm{SFT}}$ and $\pi_g$. This shows that SFT cannot outperform the teacher policy beyond this bound.
> - **Dominance of RL.** Theorem 2 then proves that the RL solution $\pi_{\mathrm{RL}}$ dominates SFT:
>   $$
>   J(\pi_{\mathrm{RL}}) \\ge\ J(\pi_{\mathrm{SFT}})
>   $$
>
> - **Implication for Rec-R1.** Together, these results provide a more rigorous justification for why RL (as instantiated by Rec-R1) can surpass the ceiling of SFT. This addresses the reviewer’s concern and clarifies the theoretical superiority of RL-based optimization in our setting.  *We then can also understand it Intuitively: SFT is fundamentally limited by imitating the teacher policy, while RL explicitly optimizes for the reward function. Therefore, if the teacher is suboptimal, RL can leverage exploration and optimization to break through this ceiling.*
>
> ---
>
> Below are the proof:
>
> **Lemma 1 (Downstream Performance Difference – KL Upper Bound)**
>
> We prove step by step.
>
> Suppose the reward is bounded:
> $$
> 0 \le f(a\mid s) \le R_{\max}.
> $$
>
> For any policy $\pi$ and the data-generating (teacher) policy $\pi_g$, define
> $$
> J(\pi) \= \ \mathbb{E}_{s\sim p(s),\,a\sim \pi(\cdot\mid s)}\big[f(a\mid s)\big].
> $$
>
> Then
>
> $$
> |J(\pi)-J(\pi_g)| \le R_{\max} \mathbb{E}_{s}\mathrm{TV}[(\pi(\cdot\mid s),\pi_g(\cdot\mid s))]
> $$
>
> $$
> \le R_{\max} \sqrt{\frac12 \kappa}
> $$
> where $\kappa$ is the expected KL divergence between $\pi_{\mathrm{SFT}}$ and $\pi_g$.
>
> *Proof:*
>
> Fix a state $s$. Denote
> $$
> P(a)=\pi_g(a\mid s), \quad Q(a)=\pi(a\mid s).
> $$
>
> Normalize rewards:
> $$
> g_s(a) = \frac{f(a\mid s)}{R_{\max}} \in [0,1].
> $$
>
> Then
>
> Δ(s) = E_{a$\sim$Q}[ f(a | s) ] - E_{a$\sim$P}[ f(a | s) ]
> = R_max * ( E_Q[g_s] - E_P[g_s] ).

---

> ### Author Response · Authors · 2025-08-21
> **Response by the Authors**
>
> By the dual representation of total variation,
> $$
> \sup_{0\le g\le 1} \big|\mathbb{E}_Q[g]-\mathbb{E}_P[g]\big| = \mathrm{TV}(Q,P).
> $$
>
> Since our $g_s$ lies in $[0,1]$, we directly have
>
> $$
> |\mathbb{E}_Q[g_s]-\mathbb{E}_P[g_s]|\\le\\mathrm{TV}(Q,P).
> $$
>
> Thus
> $$
> |\Delta(s)| \\le\ R_{\max}\ \mathrm{TV}(\pi(\cdot\mid s),\pi_g(\cdot\mid s)).
> $$
>
> Finally, take expectation over $s$ and apply Jensen’s inequality:
> $$
> |J(\pi)-J(\pi_g)| \\le\ R_{\max}\ \mathbb{E}_s[\mathrm{TV}(\pi,\pi_g)].
> $$
>
> Then, we leverage the Pinsker’s Inequality
>
> For each $s$, Pinsker gives
>
> $$
> \mathrm{TV}(\pi(\cdot\mid s),\pi_g(\cdot\mid s))
> \\le\ \sqrt{\tfrac12\,D_{\mathrm{KL}}(\pi_g\Vert \pi)}.
> $$
>
> Average over $s$ and apply Jensen again:
>
> $$
> \mathbb{E}_s[\mathrm{TV}]
> \\le\ \mathbb{E}_s\[\sqrt{\tfrac12 KL}\]
> \\le\ \sqrt{\tfrac12\\mathbb{E}_s[KL]}.
> $$
>
> Therefore,
>
> $$
> | J(π) - J(π_g) |
>   ≤ R_{max} \sqrt{( \frac12 E_s[ KL( π_g || π ) ] )}
> $$
>
> Done.
>
> $\square$
>
> Thus Lemma 1 rigorously shows: if SFT drives KL divergence to 0, then the reward difference vanishes at the rate $\sqrt{\text{KL}}$. This explains why **SFT’s ceiling is the teacher policy $\pi_g$**.
>
> **Theorem 2 (Dominance of RL Objective over SFT)**
> - Rewards are bounded: $f(a\mid s)\in[R_{\min},R_{\max}]$, width $B=R_{\max}-R_{\min}$.
> - Policy class $\Pi$ is shared by SFT and RL.
> - Objective:
>   $$
>   J(\pi) = \mathbb{E}_{s,a\sim\pi}[f(a\mid s)].
>   $$
> - Teacher policy: $\pi_g$.
>   SFT solution: $\pi_{\mathrm{SFT}}\in\Pi$. Define $\kappa$ is the expected KL divergence between $\pi_{\mathrm{SFT}}$ and $\pi_g$.
> - Optimal-in-class policy:
>   $\pi^\* \in \arg\max_{\pi\in\Pi} J(\pi)$.
>   RL solution: $\pi_{\mathrm{RL}}\in\arg\max_{\pi\in\Pi}J(\pi)$.
>   Teacher gap: $\Delta^\* = J(\pi^\*)-J(\pi_g)\ge0$.
>
> We claim: **Superiority of RL Objective over SFT**
> $$
>    J(\pi_{\mathrm{RL}})\\ge\ J(\pi_{\mathrm{SFT}}).
> $$
>
>
> **Proof**
>
> From Lemma 1,
> $$
> |J(\pi_{\mathrm{SFT}})-J(\pi_g)| \\le\ \varepsilon_{\mathrm{SFT}}.
> \tag{L1}
> $$
>
> Since $\pi_{\mathrm{RL}}$ maximizes $J$ over $\Pi$ and $\pi_{\mathrm{SFT}}\in\Pi$,
> $$
> J(\pi_{\mathrm{RL}})\\ge\ J(\pi_{\mathrm{SFT}}).
> $$
>
> Till now, **we have proved that RL policy is superior compared with SFT policy**.
>
> Done.
>
> $\square$
>
> ---
>
> We thank the reviewer again for the insightful comments. The additional analysis above clarifies the theoretical ceiling of SFT and the weak dominance of RL, showing why Rec-R1’s RL-based optimization is not only empirically effective but also theoretically justified. We will revise the paper to incorporate these discussions, which we believe will further strengthen the clarity and rigor of our contributions.

---

> ### Comment · Reviewer_Q5qx · 2025-08-26
>
> I appreciate the authors for addressing my questions and concerns.
>
> First, for the parts mentioned as revised or to be revised, I would like to see the manuscript updated with color highlighting to show clearly what has been changed. For example, in response to #6, which states "We have strengthened our theoretical analysis in the revision," it is difficult for the reviewer to understand what changes have been made.
>
> I list additional questions below:
>
> 1. It would be beneficial if the "user-centric" aspect used in the title could be more explicitly reflected in the paper's introduction and content.
>
> 2. In the authors' response #4, it would be helpful to explain further why learning co-occurrence patterns is considered a "shortcut."
>
> 3. Regarding response #5, where the authors state that their reward structure is not different from existing work (e.g., Stream-Rec), it would be good if this could be further discussed as related work in Appendix B.
>
> 4. In the final part of the proof of Lemma 1, hasn't Jensen's inequality been applied in the wrong direction?

---

> > ### Author Response · Authors · 2025-08-28
> > **Response by the Authors**
> >
> > We thank the reviewer for the constructive feedback. As suggested, we have updated the manuscript with **blue highlights** to indicate all revisions, including the additional proofs, the discussion and experiments on Rec-R1’s multi-objective reward structure, and other clarifications mentioned in our previous responses.
> >
> > 1. On “user-centric.”
> >
> > We have revised the introduction to explicitly emphasize the user-centric aspect. In particular, we now describe our focus as *“user-centric recommendation scenarios, where the key challenge is to understand language-driven inputs, capture implicit user intent, and align recommendations with user needs.”* This framing more clearly connects our work with the paper’s title.
> >
> > 2. On co-occurrence patterns as a “shortcut.
> >
> > Thanks for pointing it out. The term “shortcut” is used in a relative sense, contrasting traditional recommendation models with the LLM + RL learning paradigm. Traditional sequential recommendation models are often trained from scratch on large datasets and primarily rely on approximating patterns from large-scale data. In contrast, Rec-R1 leverages pretrained LLMs and reinforcement learning, where the model samples and evaluates trajectories, reinforcing those that align with user-intent-driven rewards. These trajectories typically involve more semantic reasoning. When the task lacks a strong logical mapping between history and future items, such reasoning becomes less effective and may even introduce noise.
> >
> > So about why we say traditional models have the "shortcut" (relatively speaking), that is because they do not need to sample long trajectories or rely on the current model parameters during training. Instead, they directly fit co-occurrence statistics in a supervised manner, which allows them to exploit overlaps in the data more easily and it is a "shortcut" (no need for reasoning or depending on current model parameters.). This is fundamentally different for the other side, where the optimization process depends on trajectory sampling and reinforcement signals, making the learning more reasoning-driven but also more challenging in tasks without clear logical mappings.
> >
> > 3. On reward structure.
> >
> > Thanks for your suggestion. We have added this discussion in the revised version of our manuscript, just as our previous response. (Appendix E.7)
> >
> > 4. On Lemma 1 and Jensen’s inequality.
> >
> > We carefully revisited the proof. The application of Jensen’s inequality is correct:
> >
> > $$
> > J(\pi) - J(\pi_g) = \mathbb{E}_{s \sim p}[\Delta(s)],
> > $$
> >
> > By using
> >
> > $$
> > \lvert \mathbb{E}[X] \rvert \le \mathbb{E}[\lvert X \rvert],
> > $$
> >
> > we have,
> >
> > $$
> > \lvert J(\pi) - J(\pi_g) \rvert
> > = \bigl\lvert \\mathbb{E}_{s \sim p}[\Delta(s)] \\bigr\rvert
> > $$
> >
> > $$
> > \le\ \mathbb{E}_{s \sim p}\\bigl[ \lvert \Delta(s) \rvert \bigr]
> > $$
> >
> > $$
> > \le\ R_{\max}\\mathbb{E}_{s \sim p}\\left[
> >     \mathrm{TV}\bigl(\pi(\cdot \mid s), \pi_g(\cdot \mid s)\bigr)
> > \right]
> > $$
> >
> > This step directly corresponds to the derivation shown in the previous response. Hence, the inequality is applied in the standard and correct direction.
> >
> > ---
> >
> > We sincerely thank the reviewer again for the constructive suggestions and careful reading of our manuscript. We have incorporated the requested clarifications and highlighted all revisions in blue in the updated version. We greatly appreciate the reviewer’s detailed feedback, which has helped us refine both the presentation and positioning of our work. We believe the updated manuscript addresses the raised concerns, and we remain open to further discussion if there are remaining questions.

---

> > > ### Comment · Reviewer_Q5qx · 2025-09-03
> > >
> > > Thank you for addressing my concerns and for clearly marking all revisions in the manuscript with blue highlighting. I have reviewed the updated version and appreciate the authors' efforts to incorporate the suggested clarifications.

---

> > > > ### Author Response · Authors · 2025-09-05
> > > > **Response by the Authors**
> > > >
> > > > We sincerely thank the reviewer for the positive feedback. We truly appreciate your recognition of our efforts in addressing the suggested clarifications. Your constructive comments have been invaluable in helping us improve the quality and clarity of our work.

---

### Author Response · Authors · 2025-10-08

Hi Reviewers,

We sincerely appreciate your thoughtful comments and valuable feedback on our manuscript. As the formal discussion period is approaching its end, we would like to kindly check whether you have any additional questions or comments regarding our work.

Over the past discussions, we have carefully considered all feedback and improved the manuscript based on your suggestions. We truly appreciate the constructive input that has helped us enhance the clarity and quality of our work.

We welcome any remaining questions you may wish to raise, and we would be glad to further clarify if needed.

Thank you once again for your engagement and time.

Best regards,
Authors

---

### Decision · Action_Editor_RHKf · 2025-10-07

**Recommendation:** Accept with minor revision

**Additional Comments:**

I only  have one minor request.

Theorem 1 (maximizing corss entropy mimizes KL divergence) is a very classic result in machine learning (https://en.wikipedia.org/wiki/Cross-entropy#Cross-entropy_minimization), I do not think it is  fair to call it a "theorem", as it makes it look as a contribution of your own. Before publication, I suggest you change the writing so that it is clearer that this is not a new result (by changing the name to "fact", or simply by explaining that it is a known result in the text).

The same is also true of Theorem 2 to a lower extent, since if i understand correctly it is a simplified version of the classic "performance difference lemma" (see eg [1, Thm 1]). I suggest the authors simply mention this in the appendix.

[1]: Schulman, John, et al. "Trust region policy optimization." International conference on machine learning. PMLR, 2015.

**Audience:**

Yes

**Audience Explanation:**

Post training of LLMs on practical tasks is an active research field in our domain.

**Claims And Evidence:**

Yes

**Claims Explanation:**

The paper introduces Rec-R1, a method to train an LLM with feedback from a recommendation system.

The method takes as in put user data, transforms it and feeds it to a recommender system, that then score it based on estimation of downstream task performance. This performance is then used as a reward signal, and the system is trained end-to-end with reinforcement learning. The method is then tested on several recommendations scenarios, and shows strong performance gains over other LLM based methods.

All the reviewers agree that the claims are accurate and properly supported, and the authors have correctly answered the main comments of the reviews. I follow the reviewers suggestion and recommend acceptance.

---

> ### Author Response · Authors · 2025-10-12
>
> Dear Action Editor,
>
> Thank you for recognizing our work and for your constructive suggestions. We have followed your suggestions and revised the manuscript accordingly. The updated camera-ready version has been uploaded. Please let us know if any further adjustments are needed. Thanks
>
> Best regards,
>
> Authors